



# How well is Rossby wave activity represented in the PRIMAVERA coupled simulations?

Paolo Ghinassi[1], Federico Fabiano[1], and Susanna Corti[1]

[1] CNR-ISAC, Bologna, Italy

**Correspondence:** Paolo Ghinassi (p.ghinassi@isac.cnr.it)

**Abstract.** This work aims to assess the performance of state of the art global climate models in representing the upper-tropospheric Rossby wave pattern in the Northern Hemisphere and over the Euro-Atlantic sector. A diagnostic based on Finite Amplitude Local Wave Activity is used as an objective metric to quantify the strength of Rossby waves in terms of Rossby wave activity. This diagnostic framework is applied to a set of coupled historical climate simulations at different horizontal
resolutions, performed in the framework of the PRIMAVERA project and compared with observations (reanalysis). At first, the spatio-temporal characteristics of Rossby wave activity in the Northern Hemisphere are examined in the multimodel mean of the whole PRIMAVERA set. When examining the spatial distribution of transient wave activity no evident improvement is found in the high resolution ensemble. On the other hand, when examining the temporal variability of wave activity, an higher resolution is beneficial in all models apart from one. In addition, when examining the Rossby wave activity time series, no
evident trends are found in the historical simulations (both at standard and high resolutions) and in the observations. Finally, the spatial distribution of Rossby wave activity is investigated in more detail focusing on the Euro-Atlantic sector, examining the wave activity pattern associated with Weather Regimes for each model. Results show a marked inter-model variability in representing the correct spatial distribution of Rossby wave activity associated with each regime pattern and an increased horizontal resolution improves the models performance only for some of the models and for some of the regimes. A positive
impact of an increased horizontal resolution is found only for the models in which both the atmospheric and oceanic resolution is changed, whereas in the models in which only the atmospheric resolution is increased a worsening model performance is detected.

## 1 Introduction

The European continent is located at the downstream end of the North Atlantic storm track. Over this region, the variability of the large scale circulation is characterised by the coexistence of low frequency, planetary Rossby waves and higher frequencies, transient eddies (Blackmon, 1976), the latter known as Rossby wave packets (RWPs, Pedlosky (1972), see Wirth et al. (2018) for a recent review). Planetary Rossby waves have a zonal wavenumber between 1 and 3 and their phase speed is slower com-





pared to RWPs. Planetary Rossby waves are forced by the orography (mountain ranges, land-sea contrast) and can be observed
in the time averaged circulation, manifesting as large meanders in the jet stream with a quasi-stationary phase (Edmon Jr et al.,
1980; Hoskins and Karoly, 1981). RWPs on the other hand arise from the conversion of the available potential energy stored
in the meridional temperature gradient found in the midlatitudes into kinetic energy through baroclinic instability (Simmons
and Hoskins, 1979; Chang and Orlanski, 1993). RWPs have a zonal wavenumber greater than 4 (a typical value in the mid-
latitudes ranges between 4 and 8) and their life cycle typically occurs on a time scale of less than 10 days (Blackmon et al.,
1984). Although some RWPs may manifest as circumglobal waves (expecially RWPs excited by teleconnections, Wallace and
Gutzler (1981); Branstator (2002)), usually their amplitude appears localised in space (Lee and Held, 1993; Wirth et al., 2018).
RWPs propagate along the sharp Potential Vorticity (PV, Hoskins et al. (1985)) gradient associated with the jet stream in the
upper troposphere. The concurring effect of all these waves with different characteristic spatial and temporal time scales is thus
responsible for the complexity of the climate over the Euro-Atlantic (EAT) sector.

One way to analyse the climate variability characteristic of the midlatitudes is to partition the atmospheric cirlulation into
Weather Regimes (WRs). In the last decade several authors analysed the ability of state of the art climate models in representing
the synoptic scale climate variability in the midlatitudes using a Weather Regime (WR) approach (Dawson et al., 2012; Cattiaux
et al., 2013; Strommen et al., 2019; Fabiano et al., 2020, 2021). WRs are recurrent and persistent circulation patterns with a
timescale which ranges from a few to several days (up to 3-4 weeks, Straus et al. (2007)). WRs can be computed using several
techniques applied to different meteorological fields (wind, geopotential height, mean sea level pressure...); for example, one of
these approaches consists in applying a clustering algorithm to the geopotential height field on a pressure surface (Michelangeli
et al., 1995; Fabiano et al., 2020), which is the general approach used in the present work, although with some small differences.
WRs appear as a series of positive and negative anomalies of geopotential height which extend along the zonal and meridional
directions. Therefore WRs can be viewed as different phases belonging to a Rossby wave train, containing both the contribution
of planetary Rossby waves and transient RWPs.

WR are of interest because they are associated to different types of weather at the surface, depending on the position of
the circulation anomalies in the upper troposphere (Robertson and Ghil, 1999; Yiou and Nogaj, 2004; Cassou et al., 2010).
This implies that the ability of climate models to correctly simulate the observed extratropical large scale circulation in the
mid and upper troposphere is thus of fundamental importance for a reliable representation of regional climate. Furthermore,
understanding how the atmospheric circulation changes in response to global warming is a prerequisite for regional climate
predictions (Corti et al., 1999; Matsueda and Palmer, 2018; Fabiano et al., 2021). Recently, it has been debated how climate
change can have an impact on the extratropical circulation in terms of changes in the jet stream position and intensity or in
terms of amplitude or phase speed of Rossby waves. In particular, three different phenomena which may induce changes in the
dynamics of the extratropical circulation in the Northern Hemisphere have been identified: the Arctic Amplification (Serreze
et al., 2009; Screen and Simmonds, 2010), the upper tropospheric warming in the tropics, related with an increased deep
convection (Li et al., 2019) and the cooling of the polar stratosphere, driven by changes in the concentration of ozone and
greenhouse gases (Randel and Wu, 1999; Ivy et al., 2016).



Francis and Vavrus (2012) hypothesized that the recently observed reduction in the thickness difference between the North Pole and the midlatitudes related with the Arctic Amplification could slow down the jet stream and thus favour large amplitude,

quasi stationary Rossby waves, associated with more persistent weather at the surface. The authors found evidence of an increasing trend in the amplitude of Rossby waves in reanalysis data and a slow down of their phase speed, estimating the wave amplitude using a geometric approach based on the displacement of a set of geopotential height contours (Francis and Vavrus, 2012, 2015). Subsequently, however, the hypothesis and results of Francis and Vavrus were questioned by other authors. The work of Barnes (2013) and Screen and Simmonds (2013), for example, demonstrated that the results of Francis and Vavrus

depended on the metric used to quantify the wave amplitude and no evidence of a wavier jet stream was found using other diagnostic methods. Barnes and Screen (2015) pointed out that the Arctic Amplification is one of the processes which may influence the jet stream variability (and thus the variability related with RWPs and WR) and that the opposite situation found in the upper troposphere (i.e. a strengthening of the meridional temperature gradient due to the warming of the upper troposphere in the tropics and polar stratospheric cooling) can, on the other hand, intensify the jet stream. This contrasting results motivate

us to use a robust diagnostic which is able to objectively identify Rossby waves, in order to perform a quantitative analysis of the spatial distribution and temporal evolution of Rossby waves during the last few decades in the observations and check whether these features are reproduced correctly in climate models or not.

Nakamura and collaborators proposed a novel theory of Finite Amplitude Local Wave Activity (LWA) which is certainly of interest for this problem (Nakamura and Zhu, 2010; Nakamura and Solomon, 2010; Huang and Nakamura, 2016). LWA in fact

is defined in terms of meridional displacement of PV on a given quasi horizontal surface (e.g. constant pressure or entropy) at each longitude, therefore it is certainly suitable to quantify the instantaneous local waviness of the atmospheric flow. LWA is able to identify Rossby waves of different wavelengths and to quantify their amplitude even when it becomes large or even finite (for example during wavebreaking or the formation of PV cutoffs). An advantage of LWA (which follows from the material conservation of PV) is that it is a conserved quantity in a frictionless and adiabatic flow, therefore it possess an exact

conservation relation (Nakamura and Solomon, 2010; Huang and Nakamura, 2016). Meanwhile, Chen et al. (2015), following the work of Nakamura and coauthors, formulated a LWA version replacing PV with geopotential height. This variant of LWA, despite being simpler to compute from data, does not satisfy an exact conservation relation as in the formulation of Nakamura and coauthors. Such geopotential height based LWA has been used by Chen et al. (2015) and Blackport and Screen (2020) to examine waviness trends in the midlatitudes, confirming no evidence of a wavier jet stream associated with an increased wave

activity in recent years.

Ghinassi et al. (2018) extended the Local Wave Activity (LWA) of Huang and Nakamura (2016) (which was originally developed in the Quasi Geostrophic framework) to the primitive equations in isentropic coordinates, defining it in terms of meridional displacement of Ertel PV on a given isentropic surface. The use of isentropic coordinates surely adds some complexity to the analysis of RWPs, however the isentropic formulation was found to be more suitable when used in the context

of predictability of RWPs (Ghinassi et al., 2018; Baumgart et al., 2019) with respect to the Quasi Geostrophic one, since the former better identifies RWPs propagating along the sharp PV gradient at the tropopause, while the latter cannot be used in the subtropics (where the QG approximation is not satisfied), where Rossby waves may originate or migrate.





The aim of this work is to assess how well the large scale circulation over Europe and the North Atlantic is represented at first in the observations (reanalysis) and in state of the art, high resolution, global climate models. We will analyse data from the PRIMAVERA project, whose goal is to investigate the impact of the horizontal resolution in representing the climate variability and related dynamical processes in climate models. Recently, Fabiano et al. (2020) investigate how the typical WRs observed over Europe are represented in the PRIMAVERA historical coupled simulations. The authors used metrics defined in the physical space (such as mean regime patterns, jet latitude distributions or blocking index) and in the regimes phase space (such as the mean WR patterns, WR significance and variance ratios) to assess the models performance. The present analysis extends the work of Fabiano et al. (2020), focusing on the variability of the upper-tropospheric large scale flow in terms of Rossby waves associated with WRs. To achieve this, we will combine the WR diagnostic of Fabiano et al. (2020) with the LWA in isentropic coordinates of Ghinassi et al. (2018), which in the present work will be used as an objective metric for waviness.

The paper is organised as follows: in section 2 we introduce and briefly describe the theory of LWA and WR and the methodology to compute them from meteorological data. In Section 3 we analyse and describe the spatio-temporal characteristics of the wintertime Rossby wave activity in the Northern Hemisphere starting from reanalysis data and in the PRIMAVERA historical coupled simulations. Then, in section 4 LWA diagnostic is applied in combination with WRs, examining the distribution of Rossby wave activity associated with each regime pattern in the PRIMAVERA simulations, using the observations as reference. Finally, section 5 is dedicated to the discussion of our results and the conclusions.

## 2 Theory and methodology

### 2.1 Dataset

We compare the following coupled climate models participating in PRIMAVERA: CMCC-CM2 (Cherchi et al., 2019), CNRM-CM6 (Voldoire et al., 2019), EC-Earth3 (Haarsma et al., 2020), ECMWF- IFS (Roberts et al., 2018), HadGEM3-GC31 (Williams et al., 2018), MPI-ESM1-2 (Gutjahr et al., 2019). The simulations are performed with various nominal resolutions ranging from 250 to 25 km. For each model, we consider a standard resolution (low-res, LR) run and one at higher resolution (high-res, HR). For ECMWF and HadGem we also consider an intermediate resolution run (MR) for both the atmosphere and the ocean. Additional information about model characteristics and their resolutions are available in Table 1. It is important to remark that in all PRIMAVERA simulations the horizontal resolution is changed with no additional tuning or adjustment of the models (HighResMIP protocol, Haarsma et al. (2020)). Note that there is a great heterogeneity amongst the model resolutions, for example the LR runs for CMCC and HadGem have a resolution of 250 km for the atmosphere, whereas the LR in ECMWF is only 50 km. Furthermore, most models increased the resolution of both the atmosphere and the ocean components, with the exception of CMCC-CM2 and MPI-ESM1, in which only the atmospheric resolution is increased. Several ensemble are produced for all models, however since we want to examine and visualise all WR patterns for all models we consider only one member per model (the first for simplicity).

In our analysis we consider the coupled historical simulations, covering the period 1979-2015 comparing them with ERA 5 reanalysis (Hersbach et al., 2020) as reference. Daily data for winter months (DJF) are used for both PRIMAVERA simulations





**Table 1.** Models used in the analysis, listed with their components (atmosphere/ocean/ice models), the atmospheric grid used for the two versions (low- and high-res), the nominal resolution and number of levels used for the atmosphere and ocean components. Note that for HadGEM-GC31 (LL, MM, HH) and ECMWF-IFS (LR, MR, HR) we also considered an intermediate resolution run.

| Model name | CMCC-CM2 | CNRM-CM6 | EC-Earth3 | ECMWF-IFS | MPI-ESM1 | HadGEM-GC31 |
|---|---|---|---|---|---|---|
| Components | CAM4, NEMO, CICE | ARPEGE, NEMO, GELATO | IFS, NEMO, LIM | IFS (43r1), NEMO, LIM2 | ECHAM6.3, MPIOM1.63, MPIOM1.63 | UM, NEMO, CICE |
| Atmos grid | 1°x1°, 0.25°x0.25° | Tl127, Tl359 | Tl255, Tl511 | Tco199, Tco199, Tco399 | T127, T255 | N96, N216, N512 |
| Atmos. nom. res. (km) | 100, 25 | 250, 50 | 100, 50 | 50, 50, 25 | 100, 50 | 250, 100, 50 |
| Atmos levels | 26 | 91 | 91 | 91 | 95 | 85 |
| Ocean nom. res. (km) | 25, 25 | 100, 25 | 100, 25 | 100, 25, 25 | 40, 40 | 100, 25, 8 |
| Ocean levels | 50 | 75 | 75 | 75 | 40 | 75 |

and reanalysis. Data used are the three dymensional horizontal wind components, temperature and geopotential height fields. All variables are retrieved on a regular latitude/longitude grid with a 2° resolution for PRIMAVERA models and reanalysis. Pressure levels are 850,700,500,250 and 100 hPa, which are the ones available in the PRIMAVERA dataset for daily data.

## 2.2 LWA

In this section at first the theory of LWA is briefly recapped. In the primitive equations in spherical coordinates, where $a$ is the Earth radius, $\lambda$ is longitude, $\phi$ is latitude, $t$ is time and with potential temperature $\theta$ as a vertical coordinate, LWA is defined as (Ghinassi et al., 2020):

$$A(\lambda,\phi,\theta,t) = -\frac{1}{\cos\phi}\int\limits_{\phi}^{\phi+\Delta\phi}(q-Q)\sigma\, a\cos\phi' d\phi'\,. \tag{1}$$

In the above definition,

$$q = \frac{f+\zeta_\theta}{\sigma} \tag{2}$$

is Ertel PV (Ertel, 1942; Hoskins et al., 1985), with $\sigma = -g^{-1}(\partial p/\partial \theta)$ denoting the isentropic layer density and $\zeta_\theta$ the vertical component of isentropic relative vorticity. $Q(\phi,t)$ represents a specific value of PV which at any time is uniquely related to a given latitude $\phi$ through

$$\iint\limits_{q>Q} dM = \iint\limits_{\phi'>\phi} dM \tag{3}$$





where $dM = \sigma dS$ is the isentropic layer mass in differential form. $\Delta\phi$ represents the meridional displacement of a PV contour $Q(\phi)$ from latitude $\phi$ and can be multivalued when a PV contours intersects a meridian multiple times. Eq. (1) states that LWA is proportional to the meridional displacement of PV contours $Q$ from their associated latitude. LWA is phase dependent and quantifies the vigour of Rossby waves in terms of their pseudo-momentum (angular momentum per unit of mass) and its physical units are ms$^{-1}$. LWA satisfies two important properties, which will be useful for the interpretation of our results.

These properties are the generalised Eliassen-Palm relation (Andrews and Mcintyre, 1976) and the nonacceleration theorem (Charney and Drazin, 1961). The first relation describes the global conservation of LWA under conservative dynamics (in absence of nonconservative processes). The second theorem states that

$$\frac{\partial}{\partial t}(\bar{A} + \bar{u}) = 0 \, , \tag{4}$$

or that the sum of $\bar{A}$ and $\bar{u}$ for conservative dynamics is locally constant (where the bar denotes some zonal averaging). This

implies that an increase (decrease) of Rossby wave activity is associated with a reduction (acceleration) of the zonal wind.

To compute LWA we start from the horizontal wind $(u, v)$ and temperature on pressure levels, then we interpolate the variables onto a selected isentropic surface following the steps described in section 2 of Ghinassi et al. (2018). In this work, in contrast with Ghinassi et al. (2018), no zonal filter is applied to LWA to remove its phase information, since in this analysis we consider time averaged fields (where the phase averaging is uninfluential) or we want to retain the phase information

of LWA when examining the WR patterns. Unfortunately, the vertical resolution due to the available pressure levels in the PRIMAVERA simulations is quite coarse in the proximity of the tropopause. This implies a weaker isentropic PV gradient at the tropopause and translates into an underestimation of the real LWA magnitude. However, since the goal of our analysis is a model intercomparison this does not affect the interpretation of our results, provided that all variables from PRIMAVERA simulations and reanalysis are retrieved on the same vertical levels.

Furthermore, LWA is partitioned into the stationary and transient components to quantify the wave activity contribution (transient vs stationary) associated with each WR. The stationary component of LWA is estimated according to Huang and Nakamura (2017), as the LWA computed from the time mean (DJF) PV field. The transient LWA component is computed as the difference between the instantaneous LWA (i.e. the LWA computed from the instantaneous PV field at each day) and the stationary component of LWA.

## 2.3  Weather Regimes

To compute weather regimes we use the Python package named "WRtool" (available at https://github.com/fedef17/WRtool DOI???) and we closely follow the methodology described in Fabiano et al. (2020). In this work, however, instead of using geopotential height to define WRs, we use the Montgomery stream function ($M \equiv c_pT + \Phi$, eq. (3.8.3) in Andrews et al. 1987, where $c_p = 1004$ J/kg is the specific heat of dry air at constant pressure and $\Phi$ is geopotential) on the 320 K isentropic surface,

to have a consistent framework with LWA, which is defined in isentropic coordinates.

We now briefly explain the algorithm to compute WR. At first, daily $M$ anomalies are calculated as deviations from the seasonal cycle obtained from the 1979-2015 time series, smoothed with a 20 days running mean. Then the first four Empirical





Orthogonal Functions (EOFs) are calculated for $M$ anomalies on the Euro-Atlantic Sector (EAT, defined on the box between 30° N and 88° N and 80° W and 40° E) for reanalysis data. The first four EOFs for ERA5 explain 54 % of the total variance
of $M$ at 320 K.

To allow the comparison between different models and the observations we choose to work with the same reference reduced phase space for all simulations, defined by the 4 leading EOFs obtained from ERA5 reanalysis. All $M$ anomalies are projected onto this reference space, obtaining time series of principal components (PCs) for reanalysis and PRIMAVERA simulation data (we will refer to the model anomalies projected onto the reanalysis reference EOFs space as pseudo-PCs).

A K-means clustering algorithm is then applied to the reanalysis PCs and model pseudo-PCs, setting the number of clusters to four, which is commonly used in the literature (Michelangeli et al., 1995; Yiou and Nogaj, 2004; Cassou, 2008). Finally, WR are attributed minimising the distance in the reference phase space to the cluster centroid for all PCs and pseudo-PCs time series. The composites of the $M$ anomalies of all the points belonging to the corresponding cluster, which are the mean WR patterns, are computed for both PRIMAVERAs and ERA5 data. The same is done for LWA, to obtain the spatial Rossby wave
activity distribution corresponding to each regime.

## 3 Wintertime Rossby wave activity in the Northern Hemisphere

The LWA diasgnostic is at first applied to the observed (i.e. using reanalysis data) northern hemispheric wintertime time averaged flow, to visualise how the circulation in the upper troposphere appears in terms of PV and LWA. Figure 1 (a) shows Ertel PV on the 320 K isentropic level for DJF computed from ERA5 data. We selected the 320 K isentropic surface since
it intersects the tropopause in the midlatitudes, which is a desirable property when diagnosing Rossby waves using LWA (Ghinassi et al. (2018), see also Appendix A). A planetary stationary wave with wavenumber 2, associated with the Pacific and North Atlantic storm tracks is clearly evident. The meridional PV gradient appears stronger in the upstream part of the two storm tracks (over Eastern Asia and the East coast of North America) and more relaxed downstream, as we proceed towards the exit region of both storm tracks (i.e. over the northwestern Pacific and over Europe, respectively). These downstream regions
are characterized by a broad PV ridge associated with an anticyclonic circulation in the time mean. The meridional PV gradient here appears weaker due to the PV mixing induced by the eddies (Novak et al., 2015) and wavebreaking is also frequent over these regions (Martius et al., 2007; Strong and Magnusdottir, 2008), often manifesting with PV streamers and PV cutoffs (Wernli and Sprenger, 2007). The total LWA (Figure 1 panel (b)) clearly identifies both storm tracks maximising over their downstream regions. This is a well known property of LWA, which tends to emphasize the mature (large amplitude) stage of the
eddies (Huang and Nakamura, 2016; Ghinassi et al., 2018). A band of LWA extends from Europe until Siberia across Eurasia, likely to be associated with decaying finite amplitude eddies penetrating into the Eurasia continent until reaching Siberia. Here, a secondary maximum of LWA, associated with a PV trough over the upstream part of the pacific storm track is found. We now partition LWA into its stationary and transient components as decribed at the end of section 2.2. Stationary LWA (Figure 1 (c)) exhibits 3 distinct maxima: two at the beginning and at the end of the Pacific storm track and a third one over the North Atlantic
reaching western Europe. These maxima of stationary LWA are associated with a couplet of PV troughs/ridges found in the







**Figure 1.** Panel (a): time mean Ertel PV for DJF on the 320 K isentropic surface (in Potential Vorticity Units (PVU), 1PVU $\equiv 10^{-6}$ kg K m$^2$ s$^{-1}$); panel (b): total (i.e. stationary and transient) LWA (colour, units ms$^{-1}$) on the 320 K isentrope for DJF. Panel (c): the same of (b) but for stationary LWA. Panel (d): the same of (b) but for transient LWA.

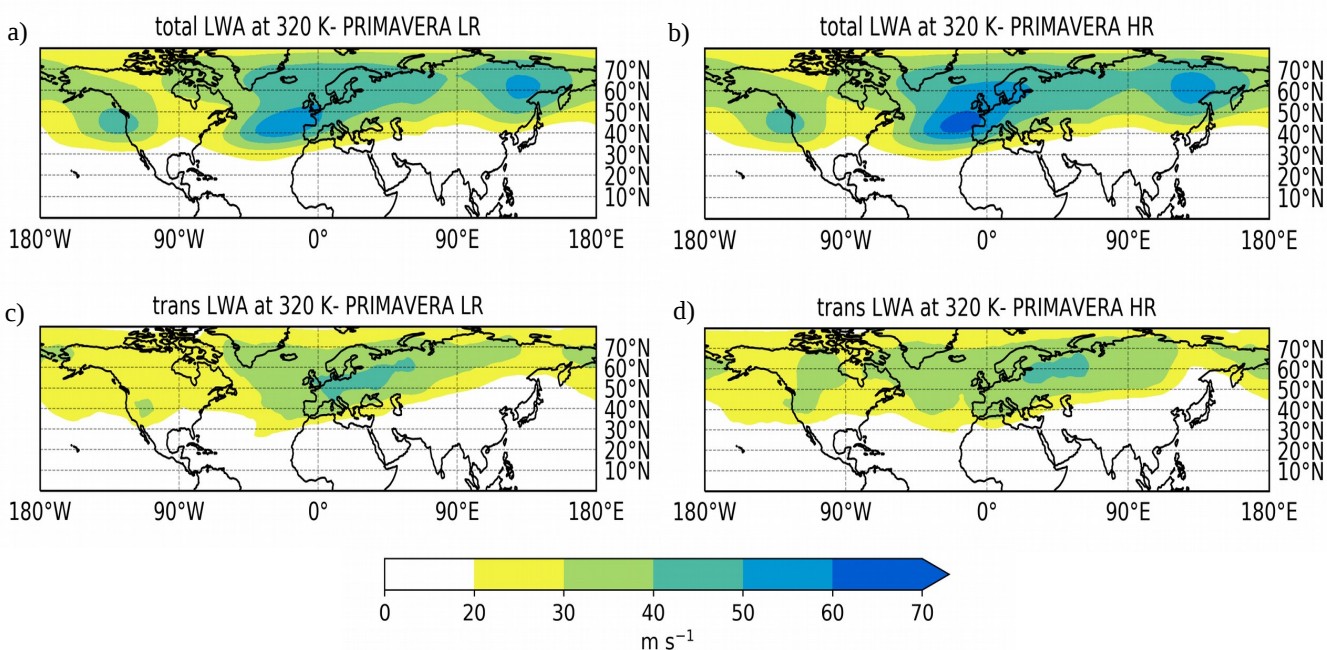

**Figure 2.** Multimodel mean of total LWA at 320 K for PRIMAVERA LR and HR (panels (a) and (b)) and for transient LWA (panels (c) and (d)).

upstream and downstream regions of both storm tracks. The LWA associated with the PV trough over the eastern part of North America does not appear in the map since its magnitude is too weak. Transient LWA (Figure 1 (d)) has a stronger magnitude compared to its stationary counterpart and it is found over a much larger portion of the domain. This implies that in the time mean picture transient eddies give the largest contribution to the total LWA in both storm tracks. Two maxima of transient LWA

are found: one over the west coast of North America, extending towards the Rocky mountains, and another one over Europe. Transient LWA in the North Atlantic storm track is much more longitudinally extended compared to the Pacific one, suggesting that transient RWPs tend to travel longer distances over the Eurasian continent, whereas at the end of the Pacific storm track the Rockies act to suppress transient Rossby wave activity immediately downstream of the mountain range. These results are in good agreement with Huang and Nakamura (2016), although the authors used the Quasi-Geostrophc formulation of LWA

and considered the vertically integrated LWA (while here we focus only on the upper troposphere).

Now we move on to investigate how the Rossby wave activity is represented in PRIMAVERA. Figure 2 shows the multimodel mean of total and transient LWA in the Northern Hemisphere for the PRIMAVERA LR and HR simulations. The multimodel mean is obtained averaging the (time averaged) LWA fields over all models. The main spatial features of the total LWA are represented correctly in both the PRIMAVERA LR and HR means. The total LWA maxima are found over the same regions

of the observations, however their magnitude is weaker. The HR helps to improve this bias especially in the North Atlantic and Europe, strengthening the total LWA maximum over this sector (compare panel (a) with (b)). When examining the plots of



transient LWA (panels (c) and (d)) instead the situation appears to sensibly differ compared to reanalysis. Overall the transient LWA magnitude in both the LR and HR ensembles is much weaker and its spatial distribution is misrepresented. In particular, in the LR mean, no LWA maximum over the downstream region of the Pacific storm track is visible and the transient LWA over Europe is shifted inland to the East compared to reanalysis. In the HR mean the situation appears to be even worse, with the LWA maximum associated with the North Atlantic storm track shifted even more downstream towards Asia. Note how this also implies that the improvement in representing the total LWA observed in the HR mean is mainly associated with a better representation of stationary LWA, especially in the EAT sector. This degrading performance in the ability of correctly simulate transient LWA in the HR mean is quite surprising and motivates us to investigate this feature in more detail. In section 4 we will examine the performance of PRIMAVERA models one by one, to reveal whether the worse performance observed in the HR ensemble is a common characteristic of all models or of only a subset of them. We will also include a WRs analysis to investigate if there are circulation patterns which are particularly sensitive to an improvement or deterioration of the Rossby wave activity pattern depending on the horizontal resolution. Lastly, after analysing the spatial distribution of LWA, we investigate the temporal behaviour of LWA. To achieve this, time series of the averaged LWA

$$\overline{A} = \frac{1}{\mathcal{D}} \int_{\mathcal{D}} A \, dS \, , \tag{5}$$

(where $\mathcal{D}$ is a certain domains and $dS = a^2 \cos\phi \, d\phi \, d\lambda$ is the area element in spherical coordinates) are produced for the midlatitudes of the NH (between 30 and 80 ° N) for both PRIMAVERAs and reanalysis.

Figure 3 shows the LWA time series for the wintertime averaged LWA in ERA 5 (black line in Fig 3 (a) and (b)) and PRIMAVERA (LR runs in in Fig 3 panel (a), HR in panel (b)). Regarding the magnitude of averaged LWA it can be seen how in general PRIMAVERA LR models tend to underestimate LWA compared to reanalysis. An increased resolution appears to be beneficial, since comparing panel (b) with (a), the HR simulations show a LWA magnitude which is closer to the observed one (particularly evident for the CMCC and CNRM models).

Figure 4 summarises the model performances in reproducing the total LWA in the NH. For each model, the lighter colour corresponds to the LR run and the darker colour to the HR. At the right end of the plot, a measure of the observed variability (black box, named "ERA5") is shown along with the average of the lowest and highest resolution versions of each model. It can be seen how the box plot confirms that an increased resolution is beneficial for almost all models (apart the MPI model) to bring the LWA magnitude closer to observations.

Regarding the temporal evolution of LWA, no significant trend is found both in the NH and in the EAT sector, with the os-cillations related to interannual variability. As we discussed in the introduction LWA is a particularly robust metric to quantify waviness since its temporal evolution can be clearly partitioned into conservative vs nonconservative propagation (eq., (15) in Ghinassi et al. (2020)). A steady LWA in the time mean picture therefore suggests a zero net effect on LWA caused by noncon-servative sources and sinks of LWA, since conservative dynamics can only rearrange the Rossby wave activity distribution. An increase or decrease of LWA with time on the other hand would imply an imbalance between sources and sinks of LWA.





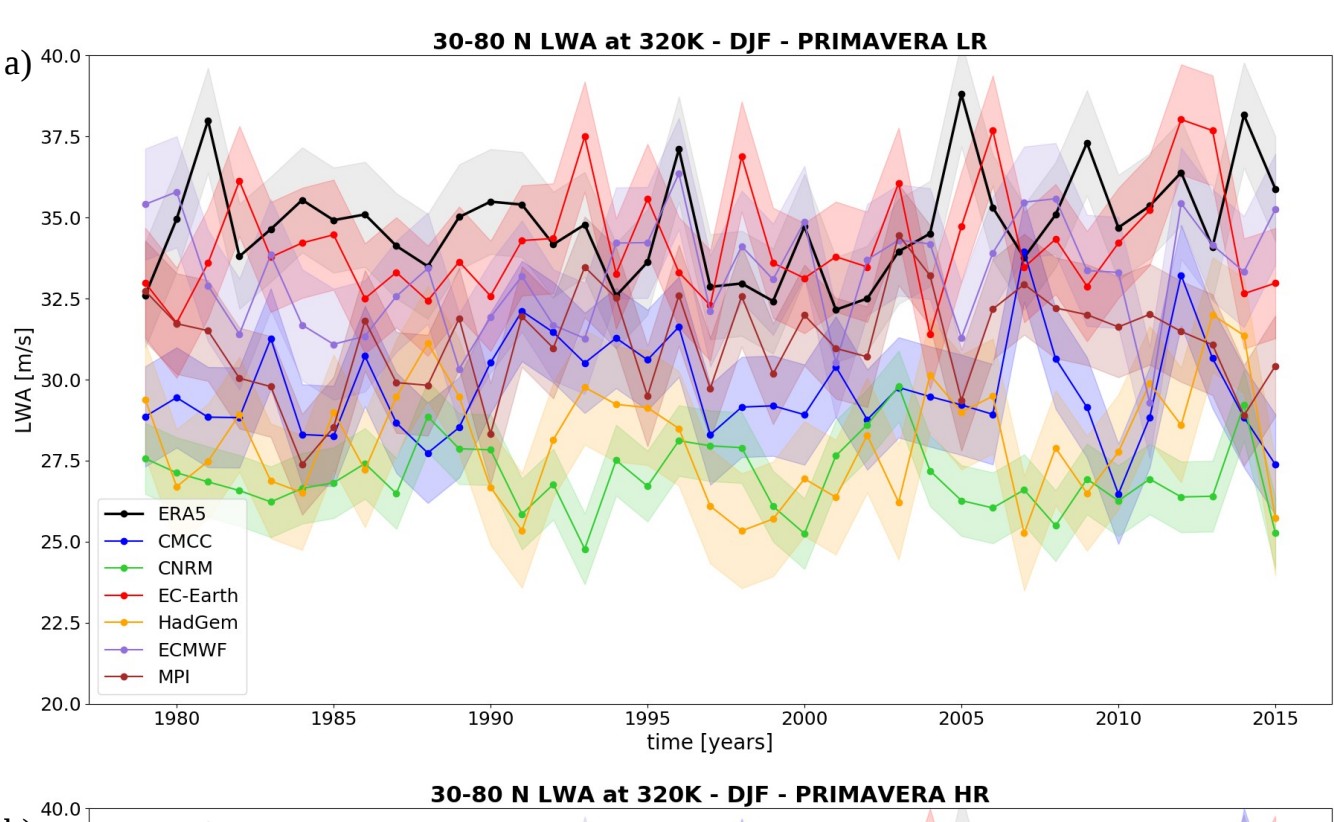

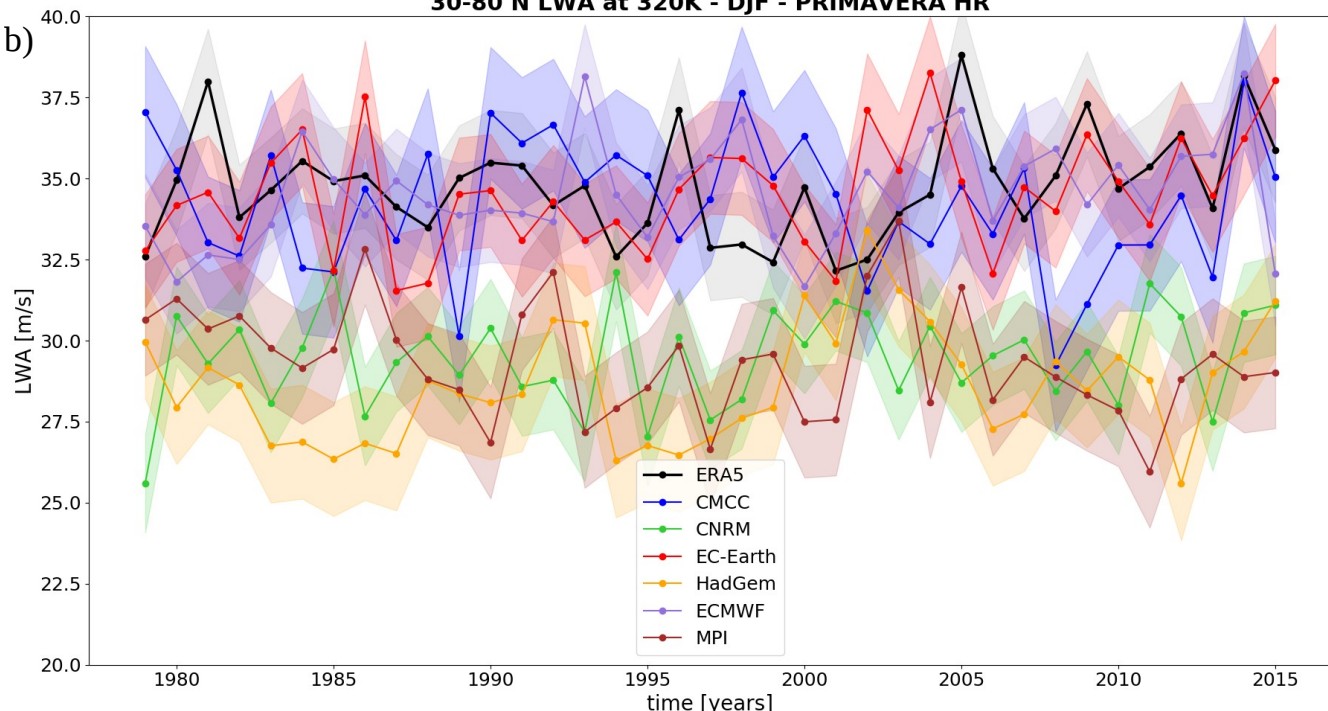

**Figure 3.** Time series of transient LWA averaged over the NH. Each dot represents the time averaged value for each winter season (DJF). Black line is ERA5, PRIMAVERA are in colour. Shading is the range between two standard deviations from the mean for each time series.

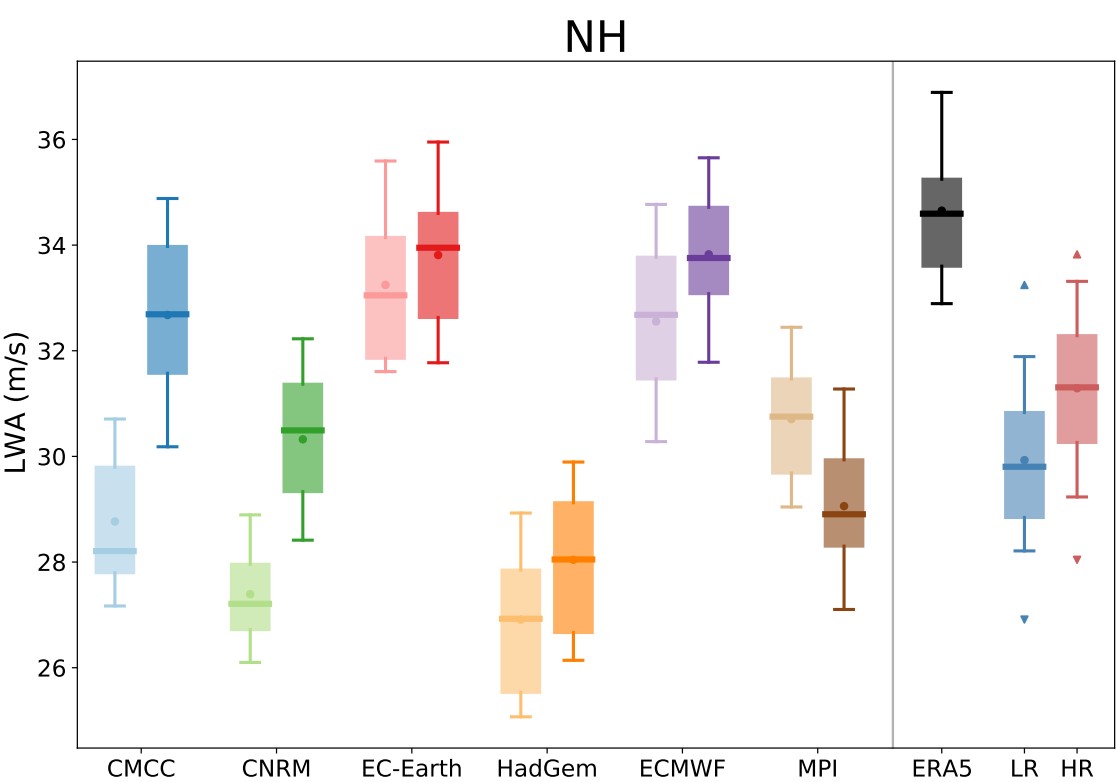

**Figure 4.** Box plot of transient LWA averaged over NH. The dots are the mean values, the horizontal line in the boxes represents the median, the boxes the first and third quartile and the bars the 10 and 90 percentiles. The left boxes are for PRIMAVERA (lighter colours are the LR runs, darker colours the HR. No intermediate resolution runs are considered here). The first (black box) on the right refers to ERA5. The other two boxes represent average quantities among all the LR and HR models and are calculated as the average of the percentiles and median over all models.





## 4   Wintertime Rossby wave activity over the Euro-Atlantic sector

We now restrict our attention over the EAT sector and analyse the wintertime large scale flow in terms of stream function anomalies and LWA associated with the four weather regimes. The four weather regimes patterns computed from reanalysis data using the Montgomery stream function on the 320 K isentrope are almost identical to the ones obtained using geopotential height at 500 hPa (Cassou, 2008; Fabiano et al., 2020). These are the positive and negative phases of the North Atlantic Oscilattion (NAO+ and NAO-, respectively), the Scandinavian blocking (short: SB) and the Atlantic Ridge (short: AR). The

only difference found lies in the frequencies of WR, which, when computed using $M$ are 28.30% for NAO+, 28.18% for SB, 22.67% for NAO-, 20.85% for AR. Compared to Fabiano et al. (2020) we found a higher NAO- frequency than for the AR (although the frequencies of the two regimes are very close). This could be due to the fact that our analysis focuses on the upper troposphere (the 320 K isentropic surface is located roughly at 300 hPa in the midlatitudes during winter, see Fig. 1 in Appendix A) and the fact that we are using a different reanalysis data set and consider a different period.

Figure 5 and 6 show the $M$ anomalies (contours) and the total and transient LWA (colour) associated with the four WR. At a fist glance it can be seen how, again, total LWA maximises over the anticyclonic phases of the regimes, while cyclonic anomalies tend to have a weaker LWA magnitude. This is particularly evident for the transient LWA component, which is very weak in correspondence of the cyclonic $M$ anomalies. Note that the stream function anomalies are very weak outside the EAT domain considered whereas a strong signal is found also outside from the EAT sector when examining LWA. In particular, a

band of LWA extending over Eurasia and another maximum over the downstream region of the Pacific storm track are found in all four WR composites. If the position of the LWA band over central Eurasia does not seem to change significantly, the location of the secondary LWA maximum over the pacific varies slightly in the four 4 WR. As can be seen comparing all panels of Figure 5 with the corresponding ones of Figure 6, such variability over the pacific is mainly linked with the transient LWA component.

We now describe in more detail the large scale circulation associated with each WR.

– NAO +: in terms of stream function this regime is characterised by a broad cyclonic vortex in the North Atlantic and a narrow band of positive anomalies over southern and central Europe. LWA maximises over this anticyclonic stream function anomaly. The cyclonic $M$ anomaly in the North Atlantic appears mainly associated with the stationary LWA component (since it is visible in the total LWA map, but appears very weak in terms of transient LWA), whereas anti-

cyclonic $M$ anomalies are mainly associated with transient LWA (see figs 2 and 3). This band of anticyclonic LWA is likely to be associated with anticyclonic Rossby wave breaking over southern Europe and the Mediterranean. LWA in the North Atlantic is very weak (transient LWA is almost suppressed), consistent with a tilted jet stream deviating to the North with the characteristic SW-NE axis (remember that LWA and $u$ are anti-correlated due to the nonacceleration theorem). Over the Pacific, a band of zonal transient LWA extends upstream of the Rockies.

– SB: LWA maximises over the wide anticyclonic stream function anomaly extending from the middle of the North Atlantic to the British isles and Scandinavia. Both stationary and transient LWA contribute to the total LWA associated with SB. The former is mainly located over the western flank of the block, whereas the latter is found more on the centre



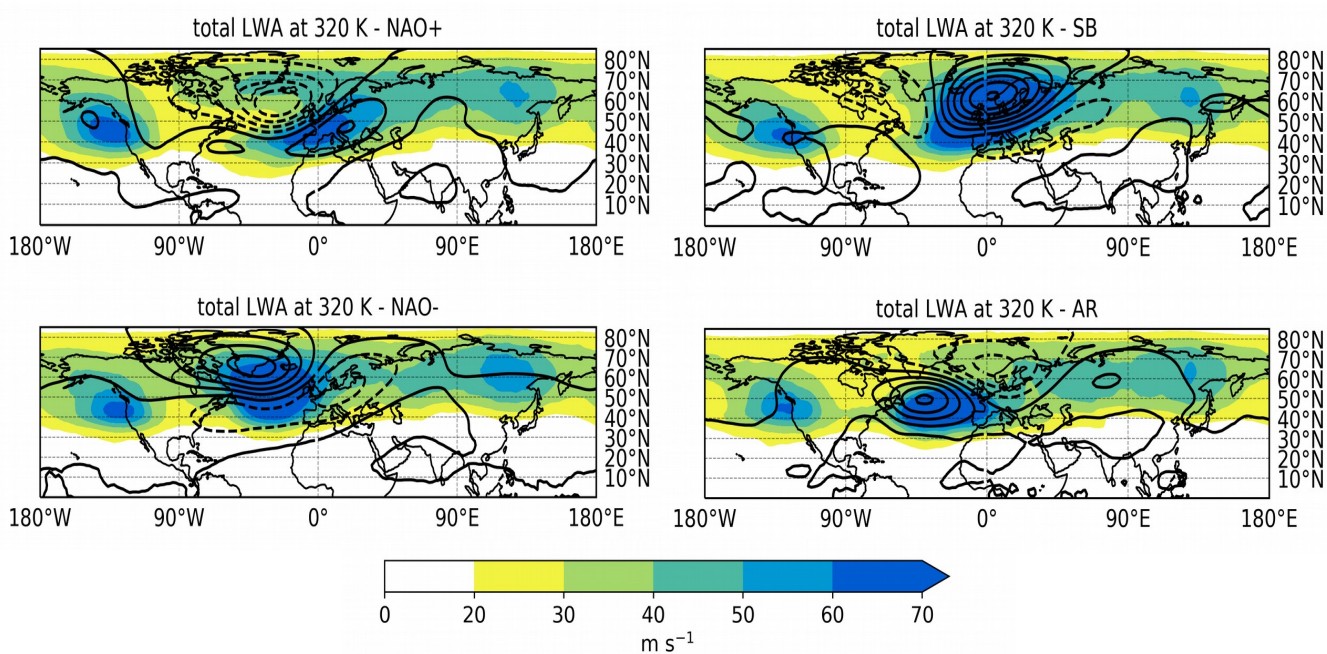

**Figure 5.** Total (i.e. stationary and transient) LWA (colour, units ms$^{-1}$) and Montgomery stream function anomalies (black contours every 100 m$^2$s$^{-2}$, dashed contours represents negative values) at 320 K associated with the four WR over the EAT sector during winter in ERA5.

and eastern flank of the structure. The high LWA values over the North Atlantic imply a very weak jet stream. A band of transient LWA is found downstream of the SB, linked with the band of negative $M$ anomaly found over the Mediter-

ranean. Over the Pacific transient LWA appears weaker compared to NAO+, with a maximum over the US portion of the Rocky mountains.

– NAO-: In terms of $M$ and LWA it is similar to a SB pattern but the whole structure is shifted to the West. A broad area of LWA is found over southern Greenland and in the middle of North Altantic. The wave activity pattern is characterised by an anticyclonic circulation on its poleward flank, cyclonic on its equatorward flank. LWA is found poleward of the

negative $M$ anomaly in the North Atlantic, consistent with a zonal jet stream displaced at southern latitudes. LWA over the Pacific extends downstream of the Rockies towards Canada and Greenland forming a "corridor" of LWA which reaches the Atlantic basin.

– AR: In terms of stream function the AR appears as a region of positive anomalies over the North Atlantic, south of 55 ° N. This large anticyclone extends in longitude from the East coast of North America to the Mediterranean across

the Atlantic. Over this ridge a maximum of LWA is found, extending from the Atlantic reaching the Mediterranean, consistent with a weaker jet over these regions. A large negative $M$ anomaly is found to the NE of the ridge, between



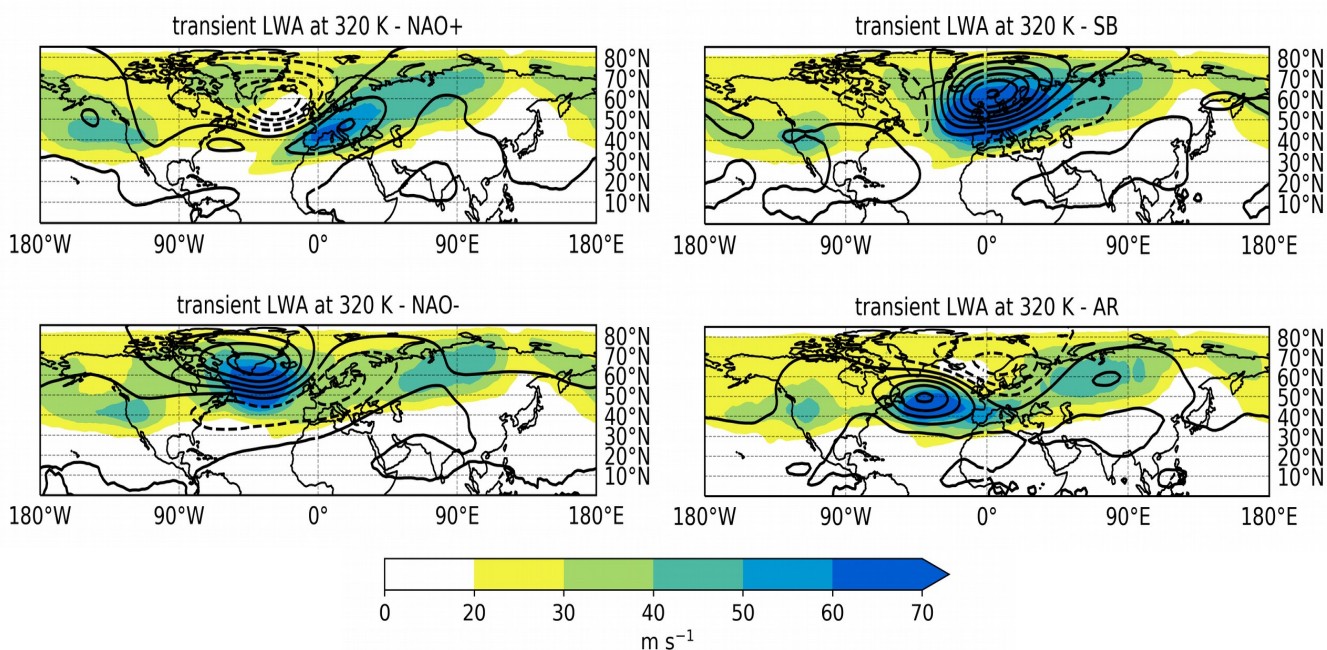

**Figure 6.** Transient LWA (colour, units ms$^{-1}$ and Montgomery stream function anomalies (black contours every 100 m$^2$s$^{-2}$, dashed contours represents negative values) at 320 K associated with the four WR over the EAT sector during winter in ERA5.

Greenland and Scandinavia. The LWA associated to this cyclonic vortex is mainly stationary (compare total and transient LWA plots for AR in figs. 5 and 6).

Now we proceed to examine how LWA associated with the four EAT WR is represented in the PRIMAVERA models,
focusing on the main differences observed when the horizontal resolution is increased. As anticipated in section 3, in our comparison we focus only on transient LWA, associated with RWPs, since it is the one that PRIMAVERA models struggle to represent correctly and no clear benefits could be observed in the HR ensemble.

Before starting our analysis we verified that the differences in the height of the selected isentropic level are negligible between reanalysis and PRIMAVERA (see Appendix A). Then, we examined the spatial pattern correlation between the mean
transient LWA pattern in each PRIMAVERA run against ERA5, which is shown in Figure 7. It can be seen how the majority of the PRIMAVERA models represents the transient LWA pattern in a satisfactory way (values of pattern correlation are larger than 0.5 apart from one case) . SB and NAO- are the regimes with the higher pattern correlation, whereas NAO+ and AR have slightly lower values on average. In some models and for some regimes, the HR simulations have a higher pattern correlation than the LR runs, suggesting that an increased resolution may improve the representation of the transient wave activity pattern
associated with WRs. The improvement of the LWA pattern correlation with resolution however is not systematic in all models. In EC-Earth for example it is almost ineffective for all regimes. Then, there are some exceptions in which the HR run has a

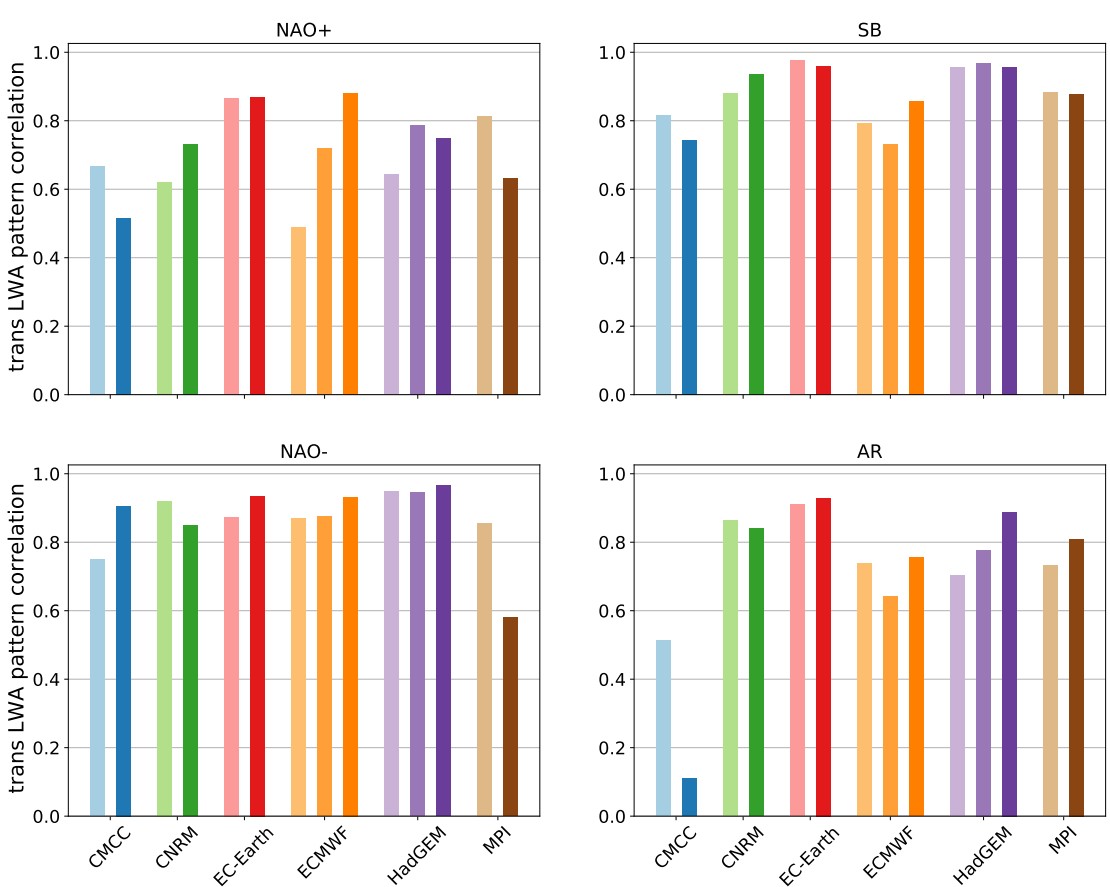

**Figure 7.** Pattern correlation of transient LWA on the 320 K isentropic surface associated with the four WR over the EAT sector during winter. Ligher colours are the LR simulations whereas darker colours are the HR ones.





lower pattern correlations than the LR, for example in the MPI model (all regimes apart from AR), in the CNRS model (NAO-and AR) and the CMCC model (all regimes apart NAO-). The CMCC HR run also fails almost completely to represent the transient LWA pattern associated with the AR.

Now that we examined the LWA pattern correlation we move on the visualisation of spatial maps of transient LWA for each regime and model. The pattern correlation in fact, despite being a concise metric to assess model performance does not provide any information about the spatial distribution of LWA in the different models. Figures 8, 9, 10 and 11 show the $M$ anomalies and transient LWA associated with NAO+, SB, NAO- and AR, respectively, for all PRIMAVERA simulations (LR and HR runs for all models, for ECMWF and HadGem we show also the intermediate resolution run). Due to the large amount of maps 325 to analyse, we will not discuss in details all of them, but instead we will summarise the salient results for each regime in the following paragraph.

- NAO+: EC-Earth is a good example on how an increased resolution is beneficial in improving the spatial LWA distribution. Despite there is almost no difference in the pattern correlation between the LR and HR over the EAT sector, transient LWA maps reveal how the tail of transient LWA which extends downstream over Eurasia is reduced in the HR run. At the 330 same time, the anticyclonic LWA over South East Europe appears slightly stronger and more localised in space. In the CMCC model instead the HR run seems to perform worse than the LR. In both resolutions transient LWA has a too weak magnitude, but the HR run fails almost completely to reproduce the anticyclonic LWA over eastern Europe. In ECMWF and HadGem the performance improves with resolution, with the LWA patterns converging towards the observed one in the runs with higher resolution. ECMWF has a too strong positive $M$ anomaly and LWA on the equatorward flank of the 335 jet, presumalby due to an overestimation of anticyclonic wave breaking during the NAO+ phase.

- SB: All models apart from CMCC do a fair job in representing the pattern associated with blocking. In particular in the ECMWF and CNRM models an increased resolution strengthen and broadens the LWA associated with blocking. The HR run of the MPI model is an exception, since in this case the $M$ and LWA pattern are better represented in the LR version. Note how the CMCC model, despite having a good LWA spatial correlation over the EAT sector, almost fails 340 to represent the Rossby wave activity pattern in the NH in both the LR and HR simulations. LWA in fact is found too shifted to the Southeast (LR simulation) or it appears distributed over a too large portion of the NH (HR).

- NAO-: almost all models succeed in reproducing the couplet of anticyclonic LWA (associated with a positive $M$ anomaly) over southern Greenland and the negative $M$ anomaly associated with suppressed transient LWA downstream over Europe. Notable exceptions are the CMCC model LR run, in which LWA extends over a too large region of NH and the 345 MPI HR model, where surprisingly the regime pattern is almost completely lost when the resolution is increased.

- AR: this is the regime where the PRIMAVERA models exhibit the higher variability in representing the spatial LWA pattern. Ec-Earth and MPI models seem to have the best performance in representing the regime pattern, both in terms of stream function and LWA distribution and magnitude. While in EC-Earth there is not a great difference in the large scale pattern between the LR and HR, in MPI the HR has a pattern which is closer to the observation, with a stronger



**Figure 8.** Transient LWA (colour, units $ms^{-1}$ and Montgomery stream function anomalies (black contours every 100 $m^2s^{-2}$, dashed contours represents negative values) at 320 K associated with NAO+ for PRIMAVERA.





**Figure 9.** Transient LWA (colour, units $ms^{-1}$ and Montgomery stream function anomalies (black contours every $100\ m^2s^{-2}$, dashed contours represents negative values) at 320 K associated with SB for PRIMAVERA.

**Figure 10.** Transient LWA (colour, units ms$^{-1}$ and Montgomery stream function anomalies (black contours every 100 m$^2$s$^{-2}$, dashed contours represents negative values) at 320 K associated with NAO- for PRIMAVERA.



**Figure 11.** Transient LWA (colour, units ms$^{-1}$ and Montgomery stream function anomalies (black contours every 100 m$^2$s$^{-2}$, dashed contours represents negative values) at 320 K associated with AR for PRIMAVERA.





negative $M$ anomaly located downstream of the ridge. HadGem (all resolutions) and CNRS correctly reproduce the AR pattern in terms of stream function, however LWA appears too weak. ECMWF on the other hand captures correctly the LWA pattern associated with the AR regime, but fails to reproduce the cyclonic $M$ anomaly located downstream of the ridge. Overall this region of cyclonic circulation located to the NE of the ridge between Greenland and Scandinavia is the feature which the majority of the models struggle to reproduce, and an increased resolution does not seem beneficial

in a systematic way in the majority of the simulations. Finally, note how the CMCC model HR model completely fails to represent the AR pattern (pattern correlation for the AR in the CMCC HR run is close to zero, see Fig. 4.

Finally, note how the LWA pattern over the downstream region of the Pacific storm track appears much weaker and smeared out compared to the reanalysis in all simulations and no clear, spatially localised secondary maximum of LWA can be identified over this region.

As we did for the whole NH, we now examine the temporal behaviour of LWA restricting our attention on the EAT sector, where WR are computed. Figures 12 shows the time series of LWA averaged over the EAT sector for LR (panel (a) and HR runs (panel (b), respectively. In the EAT sector the magnitude of averaged LWA is stronger in the HR runs but the gap between LR and HR appears less pronounced compared to the one observed in the NH, as can be seen in figure 13. The MPI model again is the only model in which the HR run exhibits less LWA than the LR. ECMWF and EC-Earth do not show a significant

increase in LWA between LR and HR, whereas HadGem, CNRM and CMCC show a significant increase in the magnitude of the averaged LWA. Note how in the CMCC and CNRS models (which showed the largest increase in the LWA magnitude with resolution both in the NH and EAT sector), the resolution of the HR run is considerably finer than the LR (refer to table 1). Also in this case no evident Rossby wave activity trend can be found in the time series both in the observations and LR and HR simulations.




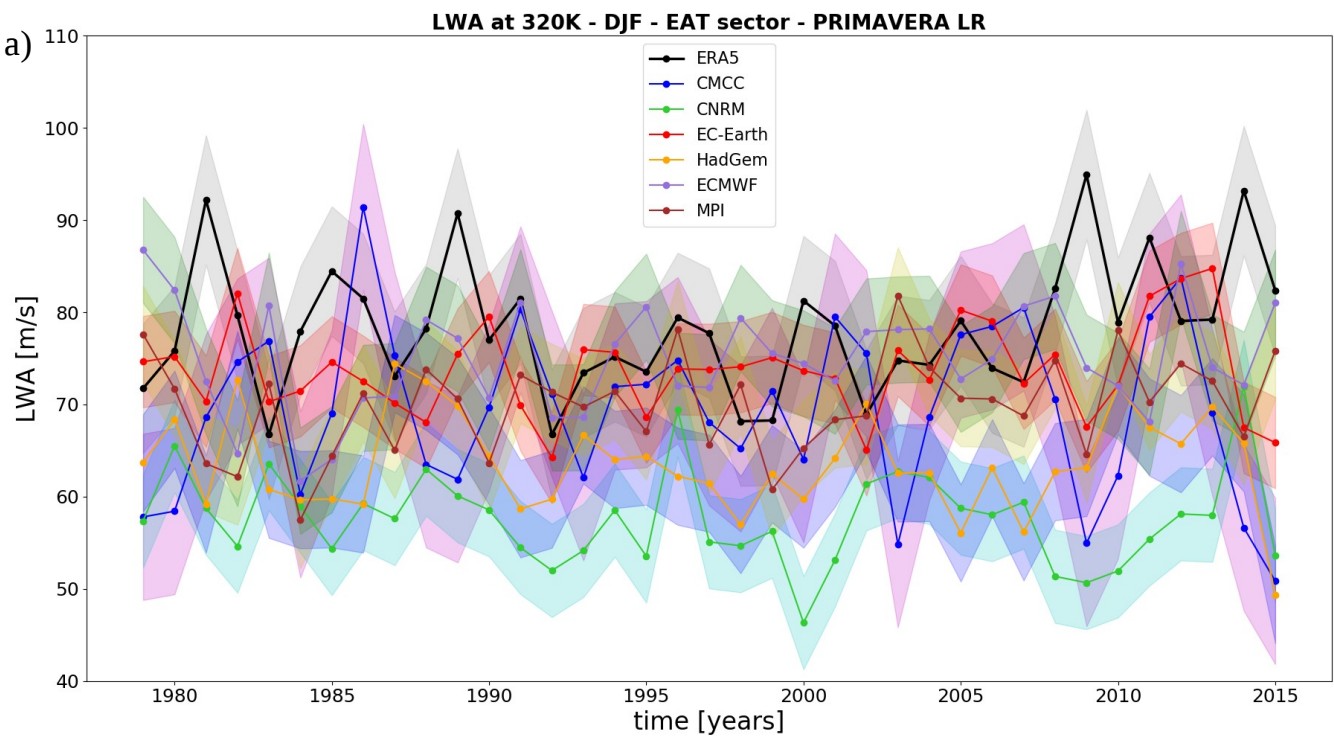

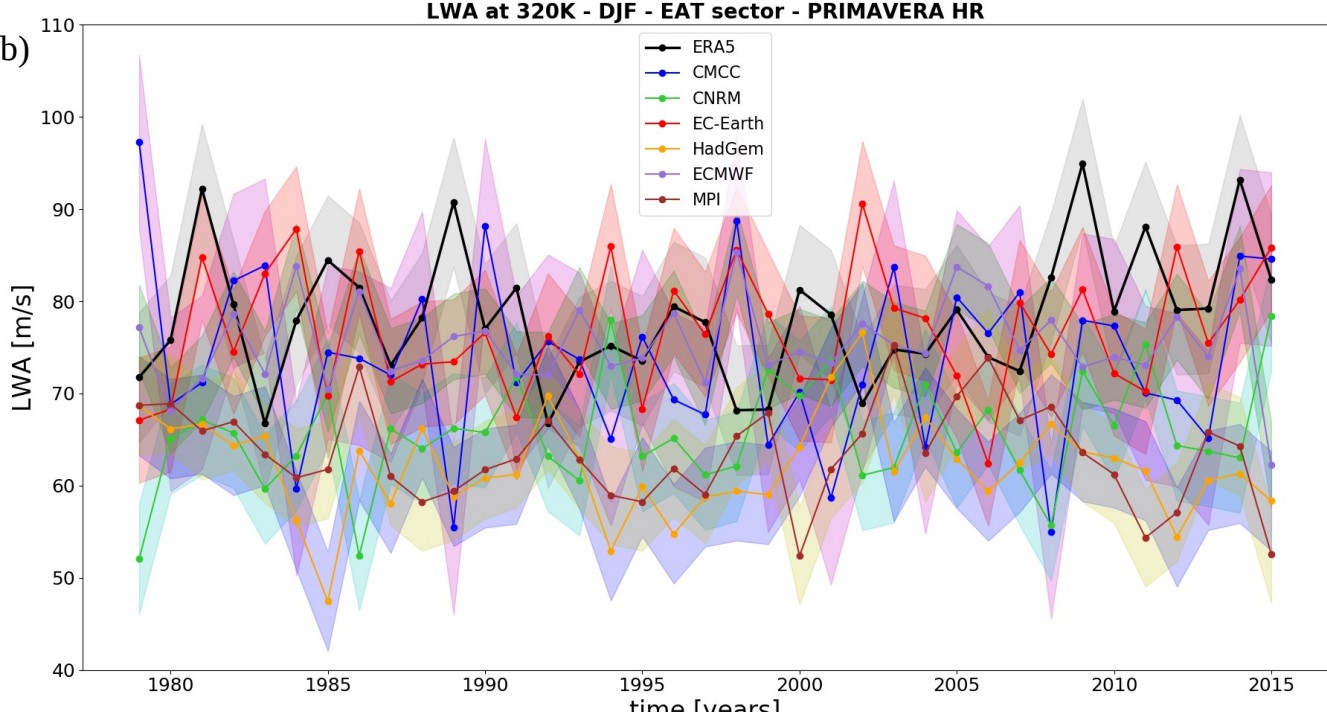

**Figure 12.** As in 9 but for the EAT sector.




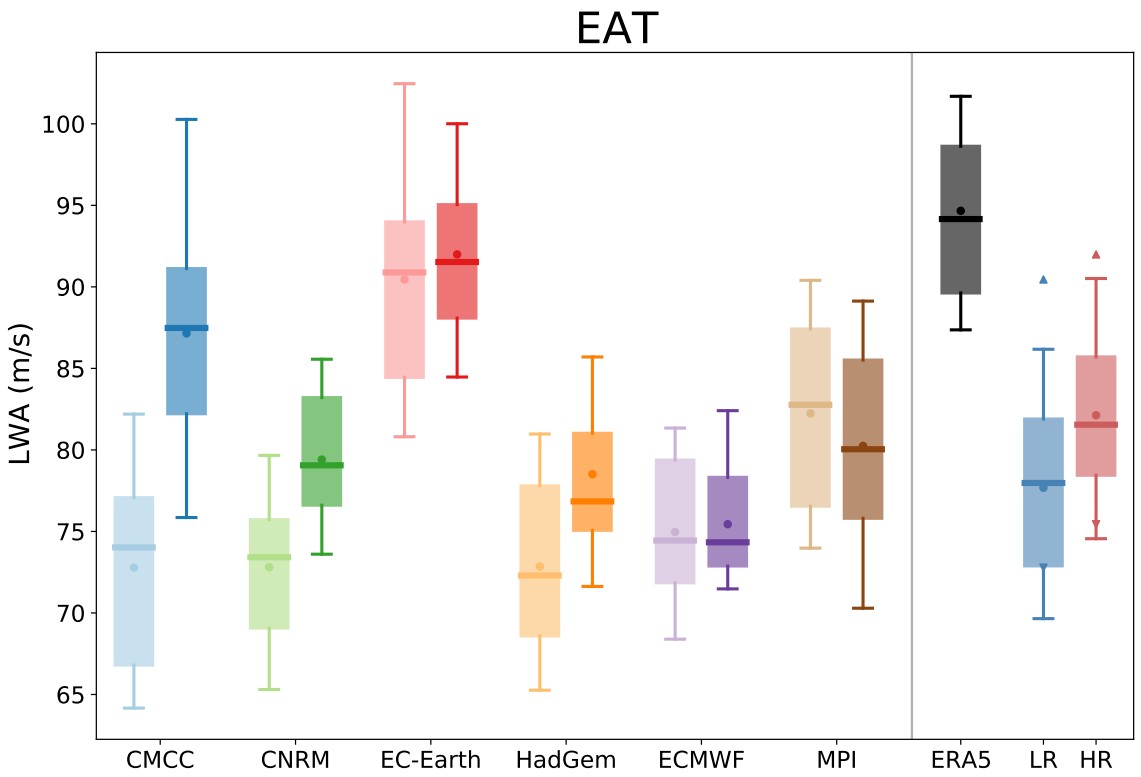

**Figure 13.** Box plot of transient LWA averaged over the EAT sector. For the explanation see caption in Fig. 4

## 5 Discussion and conclusions

In this work we have analysed the performance of state of the art climate models in representing the recurrent large scale circulation patterns associated with Rossby waves in the NH and EAT sectors during winter. In particular, the impact of an increased resolution on the representation of the large-scale atmospheric dynamics on the PRIMAVERA coupled climate simulations has been assessed, using reanalysis data (ERA5, covering the 1979-2015 period) as reference. In all models apart from two (CMCC and MPI) the horizontal resolution is increased in both the atmosphere and the ocean, whereas in the CMCC and MPI models only the atmospheric resolution is increased. Our approach combined the diagnostic for Rossby waves based on the LWA in isentropic coordinates of Ghinassi et al. (2018), to quantify their amplitude, and a weather regime analysis (following Fabiano et al. (2020)) to subsequently compute WRs over the EAT sector.

Firstly, we computed LWA for the whole NH, to analyse the wintertime large scale circulation in terms of Rossby wave activity in the reanalysis dataset. The LWA diagnostic is particularly suited to capture the large scale dynamics which char-





acterises the Pacific and North Atlantic storm tracks and identifies them in terms of Rossby wave activity associated with planetary waves and transient RWPs. The same analysis performed on the PRIMAVERA multimodel mean of the LR and HR runs revealed an improvement in the ability of the models in representing the total LWA spatial distribution in the HR, but the same was not true for transient LWA. This was attributed to a better representation of stationary LWA in the HR set; on the other hand, when examining transient LWA, we concluded that a further analysis was needed to enlighten whether the worse performance of the HR can be found in all models or it is typical of only some of them.

The temporal variability of Rossby wave activity has been addressed as well producing time series of spatially averaged transient LWA for the NH. It was evident how PRIMAVERA models tend to underestimate the magnitude of the spatially averaged LWA compared to reanalysis. In this case an increased horizontal resolution was clearly beneficial, since the magnitude of LWA in the HR simulations was closer to the reanalysis in all models apart from one (the MPI model). No trends in the evolution of LWA were found both in the observations and PRIMAVERA historical simulations (LR and HR). The evidence for no wave activity trends in the observation is in agreement with the analysis of Blackport and Screen (2020), who diagnosed waviness using a LWA variant based on geopotential height. It is worth to note that our results are not consistent with the one of Francis and Vavrus (2012, 2015), where on the opposite a recent increase in the midlatitudes waviness due to Arctic Amplification was claimed. However these authors quantified the Rossby wave amplitude using a predefined set of geopotential height isopleths, which may vary in time even due to conservative dynamics (since geopotential is not conserved). This makes it hard to tell whether the amplitude increase they have observed is related to natural variability of the geopotential height field or or rather caused by diabatic processes in the lower troposphere. On the other hand, the conservation relation of isentropic LWA (i.e. based on Ertel PV) provides a straightforward link between the rate of change of LWA due to the effect of forcings such as diabatic and other nonconservative processes. The fact that no significant LWA increase/deacrease was observed during the examined period thereby implies a net zero effect of the diabatic sources and sinks of LWA which may have an impact on the Rossby waves dynamics in the extratropics. This suggests that there is "no winner" yet in the tug-of-war between a reduced temperature gradient in the lower troposphere (related with Arctic Amplification) and an increased one in the upper levels (related with the warming of the upper troposphere in the tropics and the cooling of the lower stratosphere in polar regions), as discussed in Barnes and Screen (2015). All fluctuations observed in reanalysis and PRIMAVERA (LR and HR) therefore appeared to be associated with the intraseasonal variability of LWA.

In section 4, we restricted our attention over the EAT sector. Using the WR tool, we partitioned the LWA associated with the four WRs over the EAT sector in the observations and in the LR and HR PRIMAVERA simulations. We examined the pattern of transient LWA associated with each WR and compared it to the observations. Apart from one model (CMCC model for the AR regime) the pattern correlation showed that the large scale pattern associated with each WR was in good agreement (values larger than 0.5) with the observations. The WRs with the highest values of pattern correlations amongst the set of models are SB and NAO-. An improvement of the LWA pattern correlation in the HR simulations was not systematically observed in all models: some models showed no improvement in the pattern correlation between the LR and HR simulations and in some of the cases (CMCC and MPI) the HR runs had a lower pattern correlation for some of the regimes. If the LWA pattern correlation provides a concise metric to assess the model performance, it cannot reveal whether the error committed by



a model in reproducing the spatial distribution of LWA arises from the misrepresentation of the wave amplitude or in a shift in its phase. Examining LWA and Montgomery stream function anomalies maps we observed how LWA associated with each regime in the reanalysis appears very localised in space. LWA indeed maximises in the vicinity of the positive and negative $M$ anomalies characteristic of each regime pattern and decays to smaller values farther away. In the PRIMAVERA models instead,
the magnitude of such LWA maxima generally appears weaker compared with the observations. Furthermore some models, in spite of having high values of pattern correlation over the considered sector, fail to represent the LWA pattern in other regions of the NH. In particular, when examining the NAO+ pattern we found that the majority of the models struggles to represent the area of LWA associated with anticyclonic wavebreaking located downstream of the main cyclonic $M$ anomaly in the North Altantic. This area of anticyclonic LWA in fact appears weaker over Europe and too extended towards Eurasia, suggesting
that the models do not simulate correctly Rossby wavebreaking and the decay of RWPs over Europe and the Mediterranean, which instead continue their eastward propagation. When examining the SB, we found that two of the models which showed a substantial improvement in the LWA magnitude and pattern are ECMWF and CNRS. The HR runs of these models have a substantial increase in the horizontal resolution of both the atmosphere and the ocean. Another commonly observed feature amongst the models is the difficulty in representing the area of suppressed LWA found immediately downstream of the large
amplitude ridge which characterises the SB and AR regimes. This suggests that the models fail to reproduce the transient LWA "accumulation" into the large amplitude ridges, which is emanated downstream and does not remain trapped in the anticyclones. A similar behaviour was observed by Quinting and Vitart (2019) in the analysis of RWPs and blocking in the S2S database and was attributed to a negative RWP decay frequency and blocking frequency biases over the EAT region. A possible mechanism to explain this misrepresentation of LWA in the models is related with the tendency of the models to have
a weaker PV gradient at the tropopause (which is directly related with the magnitude of LWA) due to numerical diffusion (Gray et al., 2014; Harvey et al., 2018). Finally, there are some examples of models which almost completely fail to reproduce the observed Rossby wave activity pattern. This happens for example for the SB regime in the CMCC HR, and NAO- in MPI HR. A worse performance of the CMCC HR compared to LR in the representation of Euro-Atlantic blocking was also observed by Schiemann et al. (2020). Over the Pacific and western portion of North America a secondary LWA maximum associated
with Rossby wave activity located downstream over the EAT sector was observed in the reanalysis for all regimes. This feature was not observed in any of the PRIMAVERA simulations, suggesting that the models miss the teleconnection associated with a Rossby wave train extending from the Pacific ocean to the North Atlantic. These differences in how the models simulate the large scale circulation in the upper troposphere have implications since they are likely to be associated with errors in the circulation near the surface.

The analysis of the temporal evolution of Rossby wave activity over the EAT sector did not show substantial differences compared to the NH. The HR simulations have a LWA magnitude which is closer to reanalysis (although the gap in the LWA magnitude between HR and LR is smaller compared to the NH) apart in the MPI model. As for the NH analysis, no significant wave activity trends are visible in the EAT sector.

Concluding, the models in which the horizontal resolution is increased simultaneously in the atmosphere and in the ocean
generally show an improvement in the representation of Rossby wave activity. Notably, in the CMCC and MPI models, in





which an increased horizontal resolution degraded the models performance in simulating the spatial and temporal variability of Rossby wave activity, the resolution was increased only in the atmosphere but was left unchanged in the ocean.

Obviously our analysis focused only on the large scale circulation in the upper troposphere and does not provide information on possible biases in the dynamics at lower or higher altitudes, or happening on a spatio-temporal scale smaller than the
synoptic. In future work we will examine how the observed model biases in the upper-tropospheric Rossby wave activity are connected with surface weather and extend our diagnostic framework to future climate simulations.

*Code availability.* The WRtool package is freely available at https://doi.org/10.5281/zenodo.4590985 (Fabiano and Mavilia, 2021). The code to compute LWA from meteorological data is available from the authors upon request.

## Appendix A: Potential temperature profiles in the PRIMAVERA simulations

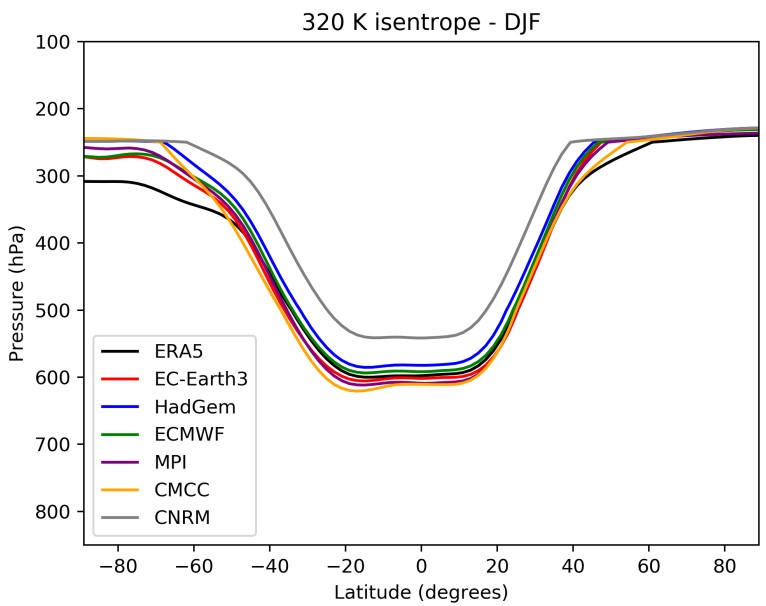

**Figure A1.** Pressure height (in hPa) of the 320 K isentropic surface (time-zonal mean for DJF). Black line is ERA5, other colours are PRIMAVERA LR simulations.

As discussed in section 2.2, LWA is partly lagrangian in latitude and altitude. This arises from the fact that isentropic surfaces evolve with time according to the variations in temperature. Since in this analysis we are comparing observations and models with different mean states we want to first verify that the considered isentrope intersects the tropopause in the midlatitudes,



which is a desirable property to identify RWPs with LWA (Ghinassi et al., 2018) and secondly that the height of such isentropic surface does not differ considerably between ERA5 and the models. Figure A1 shows the time-zonal mean of the 320 K
isentrope in the observations and PRIMAVERA LR simulations. It can be seen how the tropopause (associated with the marked change in the isentrope slope associated with a higher static stability in the stratosphere) is intercepted at around 250 hPa in the midlatitudes (around 45°N) of the NH in ERA5 and the majority of PRIMAVERAs. A notable exeption is the CNRS model in which the 320 K isentropic surface is located at an higher altitude in the troposphere, presumably due to a bias in the mean temperature. Note the bias in the height of the tropopause in the Southern Hemisphere, presumably related with a too
cold lower stratosphere during summer in the PRIMAVERA simulations (the same feature is observed in NH during JJA, not shown here). However, since our analysis focuses on the NH during winter this bias does not affect our results and will not be investigate further.

*Author contributions.* PG conducted most of the data analyses and visualizations and drafted the paper. FF performed part of the data analysis and visualizations. PG, FF and SC all commented on, organized and wrote parts of the paper.

*Competing interests.* The author declares that he has no conflict of interest.

*Acknowledgements.* The authors acknowledge support by the PRIMAVERA project of the Horizon 2020 Research Programme, funded by the European Commission under Grant Agreement 641727. The climate model simulations used in this study were performed under the PRIMAVERA project and can be accessed at the archive of the Centre for Environmental Data Analysis (CEDA). The research of HMC was supported by Natural Environment Research Council Grant number NE/P018238/1.



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
