# Peer review of "How well is Rossby wave activity represented in the PRIMAVERA coupled simulations?"

_Weather and Climate Dynamics, 2021_

## Referee Comment (RC1)

Summary and Recommendation

This work assesses the impact of model horizontal resolution on the simulation of local wave activity in the PRIMAVERA over the Northern hemisphere and Euro-Atlantic region. They classified each model to lower resolution (LR) and higher resolution (HR) versions, and then compared the ensemble mean of these two groups. They found no evident improvement of transient wave activity simulation for higher resolution. However, the conclusion based on the ensemble mean is questioned because the lower and higher bounds of model resolutions have a great heterogeneity (see the major comment). I recommend the authors to perform a major revision and reorganize the results by considering the comments listed below.

Major comment:

One of the goals of this work is to assess the impact of model resolution on the simulation of local wave activity. However, the classification of the model resolution onto HR and LR is questioned based on two reasons. On the one hand, the range of atmospheric and oceanic resolutions among different models are quite large. For example, the atmospheric resolution of LR in ECMWF-IFS is 50 km but the atmospheric resolution of HR in CNRM-CM6, EC-Earth3, HadGEM-GC31 is 50 km. On the other hand, the dynamics of LWA is dependent on spatial scales of atmospheric eddies. I thus suggest the authors to reorganize the results according to the range of atmospheric resolution: 250 km (Lower Resolution), 100 km (Standard Resolution), 50 km (Higher Resolution), 25 km (Highest Resolution) as in Scaife et al. 2019.

Minor comments:

a. The authors are also encouraged to add results based on more models in this project if the data are available. This may be helpful to reduce the impact of single model on the ensemble mean.
b. In Figs. 5-6, 8-11, I suggest to plot the wave activity anomalies instead of total field to make consistency with the stream function anomalies. In Fig 7, the spatial correlation should be correlation of the anomalous LWA between observation and models, since the high correlation in the current figure has contributions from the climatological pattern.
c. The right bracket was missing before "and Montgomery".
d. For Figs. 9-11, the captions can be changed to "As in Fig. 8 but for NAO-, SB, AR".
e. Line 167: please delete the "DOI???".
f. Line 270: please correct the phrase "in the four 4 WR".

References:
Scaife, AA, Camp, J, Comer, R, *et al*. Does increased atmospheric resolution improve seasonal climate predictions? *Atmos Sci Lett*. 2019; 20:e922. https://doi.org/10.1002/asl.922

---

## Author Response (AR1)

**Final authors response - revised version of the manuscript**

In black are the original reviewers comments to the first submitted version of the manuscript

In blue are the authors answers to the reviewers comments before providing a revised version of the manuscript.

In red are our answers according to the revised version of our manuscript. Lines refer to the revised version of the manuscript.

**Summary and Recommendation – Reviewer #1**

This work assesses the impact of model horizontal resolution on the simulation of local wave activity in the PRIMAVERA over the Northern hemisphere and Euro-Atlantic region. They classified each model to lower resolution (LR) and higher resolution (HR) versions, and then compared the ensemble mean of these two groups. They found no evident improvement of transient wave activity simulation for higher resolution. However, the conclusion based on the ensemble mean is questioned because the lower and higher bounds of model resolutions have a great heterogeneity (see the major comment). I recommend the authors to perform a major revision and reorganize the results by considering the comments listed below.

**Major comment:**

One of the goals of this work is to assess the impact of model resolution on the simulation of local wave activity. However, the classification of the model resolution onto HR and LR is questioned based on two reasons. On the one hand, the range of atmospheric and oceanic resolutions among different models are quite large. For example, the atmospheric resolution of LR in ECMWF-IFS is 50 km but the atmospheric resolution of HR in CNRM-CM6, EC-Earth3, HadGEM-GC31 is 50 km. On the other hand, the dynamics of LWA is dependent on spatial scales of atmospheric resolution: 250 km (Lower Resolution), 100 km (Standard Resolution), 50 km (Higher Resolution), 25 km (Highest Resolution) as in Scaife et al. 2019.

We acknowledge we have classified alla PRIMAVERA models in two broad categories: standard/Low resolution (LR) and High Resolution (HR), based only on the increase of the horizontal resolution of the HR compared to LR following Fabiano et al 2020 and Fabiano et al 2021. As both reviewers point out (see also comment made by the second reviewer) both the LR and HR sets contain simulation with an horizontal resolution which can vary significantly. The classification proposed by the reviewer would help to enlighten the representation of the LWA dynamics associated with atmospheric eddies with the horizontal resolution. In the revised version of the manuscript we will explore different ways of categorizing the models, possibly updating some of the figures (in particular Figure 7), and we will take into account the reviewer's concerns about the interpretation of the results.

We used the classification of models according to their atmospheric resolution as suggested by the reviewer when analysing the LWA timeseries and added an additional panel to the box plots of Figures 4 and 13. The results are discussed in the revised text at lines 254-259 for the Northern hemisphere, where we found evidence of an improvement in the representation of the spatially averaged LWA, while for the EAT sector the resolution dependence was less evident (discussion at lines 387-389).

We then applied the classification proposed by the reviewer according to the horizontal atmospheric resolution when examining the regime pattern correlation in terms of transient LWA, however no clear relation emerges from this analysis, as can be seen from the scatter plots of Figure 1 of this document.

We thank the reviewer for raising this point of discussion.

Figure 1: Scatter plot of regime pattern correlation (in terms of transient LWA) between PRIMAVERA and ERA5 vs horizontal atmospheric resolution (model colors as in Figure 7).

**Minor comments:**

a. The authors are also encouraged to add results based on more models in this project if the data are available. This may be helpful to reduce the impact of single model on the ensemble mean.

In our work we included all the PRIMAVERA models for which daily data on some vertical pressure levels of the variables used to compute LWA (horizontal wind, temperature and geopotential height for the Mongtomery streamfunction ) are available. One model (AWI) was not included since such data were not available.

b. In Figs. 5-6, 8-11, I suggest to plot the wave activity anomalies instead of total field to make consistency with the stream function anomalies.

We decided to include the full transient LWA field since we thought it would be of easier interpretation for the reader, given that LWA is not a widely used diagnostic yet. The idea is to identify the areas of high transient eddy activity with LWA and the associated circulation (cyclonic VS anticyclonic) with the Montgomery streamfunction anomalies. The stationary/climatological component has already been removed from transient LWA and it is somewhat consistent with the streamfunction anomalies, which are calculated with respect to a certain climatology. We agree that plotting transient LWA anomalies would help to identify whether the error committed by the model against reanalysis is in the amplitude of the wave or a phase shift. We will produce such LWA anomalies maps and include them in the main body of the text or in the supplementary material.

After considering several choices for Figs 8-11, we came to the conclusion that the most indicative fields to show are the transient LWA field and the transient LWA anomaly with respect to ERA5, to be able to identify model biases. We then revised the discussion of weather regimes according to these changes at lines 339-375.

In Fig 7, the spatial correlation should be correlation of the anomalous LWA between observation and models, since the high correlation in the current figure has contributions from the climatological pattern.

We computed the spatial correlation between models and observation using transient LWA, in which the stationary/climatological LWA has been removed. This is done to show how well the different models can simulate these patterns. We are not sure we completely understood the reviewer's comment in this respect.

- c. The right bracket was missing before "and Montgomery".
- d. For Figs. 9-11, the captions can be changed to "As in Fig. 8 but for NAO-, SB, AR".
- e. Line 167: please delete the "DOI???".
- f. Line 270: please correct the phrase "in the four 4 WR".

We thank the reviewer for spotting these mistakes/typos. We are going to correct them.

**References:**

Scaife, AA, Camp, J, Comer, R, *et al.* Does increased atmospheric resolution improve seasonal climate predictions? *Atmos Sci Lett.* 2019; 20:e922. https://doi.org/10.1002/asl.922

**Reviewer #2**

This study by Ghinassi et al. aims to document the representation of midlatitude Rossby wave activity in state-of-the-art climate models of the PRIMAVERA project. The authors find that overall the models reasonably well represent the climatological mean wave activity (LWA). Increasing the resolution of the models generally improves the representation of the stationary LWA but not necessarily of the transient LWA. Further, it is found that an improvement of the models can only be observed when both the oceanic and atmospheric resolution is changed. For models where only the atmospheric resolution was changed a worsening of the models' ability to represent the LWA is detected.

The study is well written and the figures are clear. However, I have some major and in my view important comments which need to be addressed. Once these comments (which may change the interpretation) have been addressed I am happy to provide a more detailed review. Just to make sure, I am generally very positive about the work. Thus, I highly encourage the authors to submit a revised version of the manuscript to WCD.

**Most important comments:**

1) Statistical significance of results: An important part of the manuscript is the discussion of the differences between reanalysis and the PRIMAVERA simulations. In my view, the discussion lacks two important aspects. 1) More quantitative information concerning the biases would be very helpful. Accordingly, I suggest to revise e.g. Fig. 2 by showing the mean LWA as contours and the differences between the PRIMAVERA simulations and reanalyses in shading. This would clearly highlight the regions associated with the most pronounced biases. A calculation and discussion of the statistical significance of the differences is missing. Thus, I would like to ask the authors to provide some information on the statistical significance. For example, a bootstrap approach with replacement would be suitable to analyse the significance of the results.

We will provide some information on the statistical significance. A bootstrap approach with replacement is definitely a way to analyse the significance of the differences against interannual variability for the multimodel plots of Figure 2.

We revised Figure 2 and added contours of the multimodel mean LWA bias (i.e. the LWA difference between the PRIMAVERA multimodel mean and reanalysis). We estimated the significance comparing the difference between the multimodel mean and the observed LWA and the standard error of the LWA seasonal averages in ERA5. We changed the text accordingly at lines 221-235.

During this analysis we found an error in the plots of transient LWA of the multimodel mean for LR and HR (we forgot to perform the time mean and only the first year was shown) and we corrected it. Fortunately the interpretation of our results remains unchanged. We thank the reviewer for these suggestions related to our Figure 2 and the significance of results, which also helped us to spot the original mistake in the figure.

2) Choice of the isentropic level: I absolutely agree that the 320 K isentropic level is a suitable choice to investigate the midlatitude LWA during Northern Hemisphere winter. However, I am wondering how this level affects any interpretations concerning RWPs along the subtropical jet. To me it is quite unexpected that no signal of LWA activity is found along the subtropical jet which stretches from Northern Africa, across the Arabian Peninsula towards India during Northern Hemisphere winter. Therefore, I encourage the authors to either include an additional higher isentropic level in their analysis, or to at least comment on possible model biases in terms of LWA along the subtropical jet.

The point raised by the reviewer is certainly of interest, however in this first work we want to focus our attention on the extratropical LWA associated with the eddy driven jet. In general, we observed that at 320 K the wintertime LWA along the subtropical jet has a much weaker magnitude compared to LWA found along the eddy driven jet (as it can be observed in our Figure 1, but see also Figure 2 of Huang and Nakamura 2017 or Figure 1 panel a of Nakamura and Solomon 2011 part II). This is likely to be related with a weaker PV gradient associated with the subtropical jet compared to the PV gradient in the midlatitudes. Considering an additional higher isentropic level to analyse subtropical LWA we believe would add too much complexity to the present work and move the focus away from the mid-latitudes.

3) Classification into HR and LR: A major goal of the study is to investigate the impact of model resolution on the LWA. To this regard, the models are classified into LR and HR. However, in its current form the classification is questionable since LR and HR actually include model runs with the same atmospheric resolution. For example, the CNRM-CM6 with 50 km is classified as LR whereas the ECMWF-IFS with 50 km is classified as HR. Accordingly, I suggest to reorganize the classification so that each of them only contains models with a similar range of atmospheric resolution. In the same way, it would be intriguing to classify the simulation based on the ocean resolution (100 km vs 25 km). I would leave the final decision concerning this latter aspect to the authors.

As we discussed in the reply to reviewer 1, in the revised version of the manuscript we will explore different ways of categorizing the models, and we will take into account the reviewers' concerns about the interpretation of the results.

This point has been raised by reviewer #1 as well, please see the discussion in the answer to reviewer 1.

4) WR identification: The authors state that "to allow the comparison between different models and the observations we choose to work with the same reference reduced phase space for all simulations, defined by the 4 leading EOFs obtained from ERA5 reanalysis." Accordingly, the anomalies from the models are projected onto the reference space. Though I understand the reasoning behind, the reader is left wondering on how potential model biases affect the projection onto the reference space. Is any

bias correction of M performed prior to the projection? This important information needs to be included and I suggest the authors to perform a bias correction prior to the regime identification.

For each model, the climatological mean field of M is removed before projecting on the reference space. So any mean state bias of the models does not affect the projection. As discussed in Sect. 3.1 of Fabiano et al. (2020), this step of the procedure is crucial for comparing results from different models/simulations. We will provide more detailed comments on this in the revised text.

The discussion about any eventual model bias in the mean state affecting the projection onto the reference space of reanalysis has been clarified and expanded at lines 178-183.

**Minor comments:**

I. 40: Better write (e.g., wind, geopotential height, mean sea level pressure) instead of (wind, geopotential height, mean sea level pressure...)

I. 225: This shift is consistent with Quinting and Vitart (2019) who found the same behaviour in models of the S2S reforecast data base.

I. 248: The absence of a significant trend is consistent with Souders et al. 2014 who also investigated trends in RWP frequency, activity, and amplitude.

We thank the reviewer for the suggestions and we will include these citations, but probably in the conclusions where we comment on our results in the context of literature.

We have included the references in our conclusion section at line 418 (Souders et al. 2014) and at line 458 (Quinting and Vitart 2019).

**How did you estimate the significance of the trend?**

In Figures 3 and 12 we plotted the transient LWA (coloured lines) plus or minus a standard deviation from the mean of each time series (shading). We did not perform a significance test to verify the presence/absence of trends but our conclusion was made analysing time series (with the shading/standard deviation representing the interannual variability) by eye. We will make this statement more rigorous in the revised version of the paper (e.g. performing a significance test).

We performed a Mann-Kendall test on the timeseries of transient LWA and we found that in ERA5 and the whole LR set the trend is not significant. In two of the HR simulations (EC-Earth and ECMWF) the trend was significant, with a p value <0.05. A typical magnitude of the LWA trend found in the PRIMAVERA simulations and observation is  $10^{-2}$  m/s per year. We discussed this at lines 260-265. We thank the reviewer for these suggestions.

I. 287: The relation between transient RWPs and blocking found in this study is consistent with the results of Altenhoff et al. (2008) and Quinting and Vitart (2019).

Thank you for suggesting these studies.

We have included the references suggested by the reviewer in our conclusion section at line 457 and 458.

I. 365: Again, how is the significance of the trend determined? And can you actually quantify the magnitude of the trend?

Here we used the word "significant" as "substantial", but no significance test has been performed. We will perform a significance test and add some quantitative comments in the revised version.

As for the Northern Hemisphere we performed a Mann-Kendall test for significance on the timeseries of the EAT sector. See discussion at lines 389-391.

**References:**

Souders, M. B., Colle, B. A., & Chang, E. K. M. (2014). The Climatology and Characteristics of Rossby Wave Packets Using a Feature-Based Tracking Technique, Monthly Weather Review, 142(10), 3528-3548.

Altenhoff, A. M., Martius, O., Croci-Maspoli, M., Schwierz, C. & Davies, H. C. (2008) Linkage of atmospheric blocks and synoptic-scale Rossby waves: a climatological analysis, Tellus A: Dynamic Meteorology and Oceanography, 60:5, 1053-1063

**References:**

Huang, C. S. Y., and Nakamura, N. (2017), Local wave activity budgets of the wintertime Northern Hemisphere: Implication for the Pacific and Atlantic storm tracks, *Geophys. Res. Lett.*, 44, 5673–5682, doi:10.1002/2017GL073760.

Nakamura, N., & Solomon, A. (2011). Finite-Amplitude Wave Activity and Mean Flow Adjustments in the Atmospheric General Circulation. Part II: Analysis in the Isentropic Coordinate, Journal of the Atmospheric Sciences, 68(11), 2783-2799, DOI: https://doi.org/10.1175/2011JAS3685.1

---

## Author Response (AR2)

We would like to thank the two reviewers for their comments and suggestions to our revised manuscript. We now answer their comments, before resubmitting a newly revised version of the manuscript. The original reviewers' comments are in black, while our replies are in blue.

**Reviewer #1:**

The authors have answered most of my questions and the manuscript has been improved significantly. However, the author did not adequately address my minor concern b. When analyzing the four weather regimes, it is better to plot the composite anomalies of transient LWA instead of total field of transient LWA because the former will help to focus on the LWA anomalies in the key region related to each regime, and thus the spatial pattern will not be affected by other regions (e.g., the high correlation over North Pacific). I thus recommend the authors to perform a minor revision by considering the comments listed below:

a) In Figs. 5-6, 8-11, I still suggest to show the composite maps of the transient LWA anomaly instead of the total transient LWA. Here I want to clarify that the composited transient LWA anomaly is calculated as the composites of total transient LWA in each weather regime minus the climatology of transient LWA in all days. It is important to note that the climatology of transient LWA is different from the stationary LWA (Huang and Nakamura 2017).

b) Similarly, in Fig. 7, it is better to show the spatial correlation of anomalous transient LWA between model and observation. It is interesting to see whether the spatial correlation of the anomalous transient LWA between model and reanalysis will become higher under higher model resolution.

We would like to thank the reviewer for suggesting this alternative analysis considering the anomalous LWA instead of the full field. We have done the suggested analysis producing analogues of Fig. 6,7 and Figs. 8-11 in terms of anomalous transient LWA. However we decided to keep these plots and their discussion as supplementary material to our manuscript. The main reason behind this is that the LWA theory in the primitive equations in isentropic coordinates in combination with Weather Regimes to our knowledge has not been applied yet to a climatological analysis involving reanalysis and several models. Therefore, in this first article we prefer to show the full LWA field (and not the anomalies) for both reanalysis and PRIMAVERA as reference for future work.
Another reason is related with the fact that from the full transient LWA field the regions of weak zonal wind can be deduced for each regime due to the non acceleration theorem, and this would not be straightforward when considering LWA anomalies. We refer to the supplementary material, where the analysis of WR in terms of anomalous LWA is presented and discussed at line 388-390:
"In addition to the analysis of WR in terms of transient LWA we repeated our approach but considering the transient LWA anomaly (i.e. the transient LWA in each WR minus the climatology of transient LWA for DJF) to exclude the model biases in the mean state. The results are presented in the supplementary material."

and in the conclusions at lines 492-93:

"The same analysis but in terms of anomalous transient LWA, which can be found in the supplementary material, also confirmed the results discussed above."

**Reviewer #2:**

This manuscript by Ghinassi et al. assesses the representation of Rossby wave activity in PRIMAVERA simulations and its sensitivity to model resolution. The authors addressed my previous comments and the manuscript has greatly improved. I still identified some minor issues which should be addressed prior to publication. In particular, the discussion of Figures 8-11 is difficult to follow since references to the individual figure panels are not provided. Such an addition would help the reader tremendously. After these comments have been addressed I recommend the study to be published in WCD.

**Minor comments:**

l. 3: In its classical physical definition the amplitude of a wave can be quantified. Is that what "strength" is referring to? Please consider to write "amplitude" instead and use this terminology consistently. Further, I think that "in terms of Rossby wave activity" is redundant since it is already being mentioned that you are using a diagnostic based on finite amplitude local wave activity. In my opinion, the sentence could end with "... of Rossby waves."

The reviewer is right, the use of strength can create confusion so we changed it to amplitude to have a consistent terminology for all the manuscript.

l. 43: Is it on purpose that the authors are using the term "Rossby wave train" here?

Yes, we used "Rossby wave train" to denote a longitudinally extended series of ridges and troughs containing both contributions from planetary, quasi stationary waves and transient wave packets.

l. 48: "thus" is somewhat redundant since the authors already state at the beginning of the sentence "this implies".

We eliminated it, thanks for the suggestion.

l. 69: A critical reader may wonder why your diagnostic is more robust than previous diagnostics. Providing a brief explanation concerning this aspect would certainly strengthen your argument. It could be sufficient to state that evidence for your statement is provided in the following (l. 72ff).

We added "This contrasting results motivate us to use a robust diagnostic based on Finite Amplitude Local Wave Activity (LWA), which is able to objectively identify Rossby waves, as we will discuss in the following paragraph." at lines 68-70.

l. 92: I think this sentence is a bit confusing. Are you really assessing how well the large scale circulation over Europe and the North Atlantic is represented in observations? What would then be your reference you are comparing to. Please revise the sentence if necessary.

We clarified this point and rephrased to "The aim of this work is to assess how well the large scale circulation over Europe and the North Atlantic is represented in state of the art, high resolution, global climate models, using observations (reanalysis) as reference." at lines 92-93.

l. 143: Please insert space between m and s^{-1}. E.g., in Latex through m\,s^{-1}.

Thank you for spotting this, we corrected it.

l. 154: This explanation is very helpful. Please consider to provide this earlier in the manuscript, e.g., when first introducing vertical gradients (e.g., l. 135) since I assume that they are heavily affected by the coarse vertical resolution.

We thank the reviewer's suggestion, however we prefer to keep the two statements separated since the sentence at line 135 refers to the theoretical aspects of LWA, while the discussion at lines 154-158 refers to our methodology and the availability of data.

l. 185: Quite often weather regimes are defined as circulation patterns that persist for several consecutive days. Are you using any minimum persistence criterion in your work or is it possible that weather regimes sometimes persist for a single day only? Please explain.

Thanks for the comment. We do not adopt here a persistence criterion for defining the regimes, so all days in the timeseries are considered in the clustering and regimes can persist even for a single day. This is in line with the analysis in Fabiano et al. (2020).

l. 197: Are you meaning the northeastern instead of the northwestern Pacific? Actually, the storm track is quite active over the northwestern Pacific at least in terms of cyclone activity.

Yes, we meant northeastern Pacific, since we are referring to the regions in which the meridional PV gradient appears weaker.

l. 201: This sentence is slightly confusing. I would rather state that "The local LWA clearly maximizes at the downstream end of the storm tracks". Compared to the classical definition of storm tracks, e.g., Chang et al. 2002, the maxima in LWA are much further east than the maxima in cyclone activity or geopotential height standard deviation.

We changed our sentence according to the reviewer's suggestion.

l. 254: This is an important result! Based on this I was wondering if the authors would like to reconsider the title of their paper. In its current form the title gives the impression of a pure verification study. However, seeing these results there is an important message about the model resolution to adequately represent Rossby waves. Perhaps asking "what resolution is needed to better represent Rossby waves in climate models?" could increase the visibility of the study. At least from the hemispheric perspective there is a clear result. For Europe, however, things seem to be much more unclear. I would leave the final decision about the title of the study to the authors.

We thank the reviewer for this nice suggestion, however we prefer to keep the title as it is.

l. 324: This statement is confusing. Figure 4b clearly shows the benefit of the HR. Am I missing something important here? Please clarify.

The reviewer is right, our previous statement was confusing. We referred to the fact that in Figure 2 (and 13) the impact of HR is not very clear over the EAT sector. We have changed the sentence at lines 323-325 with:

"As anticipated in Section~3, in our comparison we focus only on transient LWA associated with RWPs, since the benefit of a higher resolution are less evident in the LWA distribution of the multimodel mean (compare Fig.2 (c) and (d))."

l. 329: How is this threshold of 0.5 chosen? In weather forecasts, a pattern correlation (anomaly correlation coefficient) of less than 0.6 is considered to be useless. In this study it is the pattern correlation of the climatological mean. Would one not expect to see considerably higher correlations?

We are not aware of other studies which computed pattern correlations using LWA as a variable therefore we chose this value pretty much arbitrarily.

Such lower values of pattern correlation compared to anomaly correlation coefficient computed using geopotential height may be due to the fact that LWA is a highly derived quantity and its field it's not as smooth as geopotential height. We are applying here a very challenging diagnostic for models, since this requires both that the regimes are well represented (as in Fig. 3 of Fabiano et al. 2020), and that additionally the LWA composites for each regime are well represented. This is why we did not expect very high correlations here. In addition our correlation is a spatial correlation of a climatological mean which does not involve any time dependence.
With that said, there are many models that have much better performances (above 0.8), so the choice of the 0.5 threshold has been probably too conservative and we changed it to 0.6.

We modified our sentence at lines 328-331 in:

"It can be seen how the majority of the PRIMAVERA models represents the transient LWA pattern in a satisfactory way, with values of pattern correlation larger than 0.6 (apart from the CMCC model for the AR regime and NAO+ in the HR), with some of the models having a correlation larger than 0.8 (the best

model in this sense is EC-Earth which has a correlation coefficient larger than 0.8 for all four WRs)."

l. 344: Given the many panels in each Figure, please label these (a, b, c etc.). This would help the reader enormously.

We labelled the panels of Figs.8-11 and now we refer to the labelling in the discussion of WRs in PRIMAVERA at lines 347-384. Thanks for this suggestion.

l. 366: Remove double "the".

Thank you for spotting this, we removed it.

l. 405 and elsewhere: Why are the authors switching to past tense? Please check carefully for consistent tense in this last section.

We removed the inconsistencies in the tenses using the present tense. Thank you.

References:
Chang, E. K. M., Lee, S., & Swanson, K. L. (2002). Storm Track Dynamics,
Journal of Climate, 15(16), 2163-2183.

References:

Fabiano, F., Christensen, H.M., Strommen, K. *et al.* Euro-Atlantic weather Regimes in the PRIMAVERA coupled climate simulations: impact of resolution and mean state biases on model performance. *Clim Dyn* **54,** 5031–5048 (2020). https://doi.org/10.1007/s00382-020-05271-w

---

## Author Response (AR3)

We want to thank the Editor for reading our last version of the manuscript and noting some mistakes which we have corrected. We now upload the finalized version of our manuscript and supplementary material.

**Comments to the author**:

Thank you for your new revised paper, which accurately takes into account the last points raised by the referees. I would like to mention the following two points I have noticed:

- The black contours in Fig.1 of the supplementary material and Fig.6 of the paper are not exactly the same while the caption says it is the same. I wonder if in Fig6 one of the contour is zero while in Fig.1 the zero contour is omitted. Also it seems that in Fig.1 not all the contours are shown (also compared to Fig.6).

REPLY: You are right. The stream function contours shown in Fig.1 of the supplementary material are only $\pm$ 500, $\pm$ 1000, and $\pm$ 1500 $m^2 s^{-2}$ and not every 100 $m^2 s^{-2}$ as stated in the label. We decided to plot the same contours in Fig.5 and 6 of the manuscript to reduce the contours density in these figures and changed the labels accordingly. Thank you for spotting this mistake.

- Line 324: I think the word "TRANSIENT" is missing in the new sentence: "As anticipated in Section 3, in our comparison we focus only on transient LWA associated with RWPs, since the benefit of a higher resolution are less evident in the TRANSIENT LWA distribution of the multimodel mean (compare Fig. 2 (c) and (d))"

REPLY: true, we have added it.